



# Three-dimensional climatology, trends and meteorological drivers of global and regional tropospheric type-dependent aerosols: Insights from 13 years (2007–2019) of CALIOP observations

Ke Gui[1], Huizheng Che[1*], Yu Zheng[1], Hujia Zhao[2], Wenrui Yao[1], Lei Li[1], Lei Zhang[1], Hong Wang[1],

Yaqiang Wang[1], Xiaoye Zhang[1]

[1]State Key Laboratory of Severe Weather & Key Laboratory of Atmospheric
Chemistry of CMA, Chinese Academy of Meteorological Sciences, Beijing, 100081, China
[2]Institute of Atmospheric Environment, China Meteorological Administration, Shenyang, 110166, China

*Correspondence to*: Huizheng Che (chehz@cma.gov.cn)

**Abstract.** Globally gridded aerosol extinction data from the Cloud–Aerosol Lidar with Orthogonal Polarization (CALIOP) during 2007–2019 are utilized to investigate the three-dimensional (3D) climatological distribution of tropospheric type-dependent aerosols, and to identify the trends in column aerosol optical depth (AOD), partitioned within different altitude regimes, and their meteorological drivers. Using detection samples of layer aerosols, we also yield a 3D distribution of the

frequency-of-occurrence (FoO) of aerosol sub-types classified by CALIOP. The results show that the aerosol extinction coefficient (AEC) shows contrasting vertical distribution patterns over land and ocean, with the former possessing significant geographical dependence, while the enhancement of AEC in the latter is mainly located below 1 km. The vertical structures of the type-dependent AECs, however, are strongly dependent on altitude. When the total AOD (TAOD) is partitioned into the planetary boundary layer (PBL) and the free troposphere (FT), results demonstrate that the PBL and FT contribute 61.86%

and 38.13%, respectively, of the global tropospheric TAOD averaged over daytime and nighttime. Yet, this CALIOP-based partitioning of the different aerosol sub-types in the PBL and FT varies significantly. Among all 12 typical regions of interest analyzed, more than 50% of TAOD is located in the lower troposphere (0–2 km), while the contribution is less than 2% above 6 km. In global average terms, we found the aerosol FoO averaged over all layers is 4.45%, with the largest contribution from 'clean marine' (1.79%) and the smallest from 'clean continental' (0.05%). Overall, the FoO vertical structures of the aerosol

layer exhibit a distribution pattern similar to that of AEC. The resulting trend analyses show that CALIOP accurately captures significant regional anomalies in TAOD, as observed in other satellite measurements and aerosol reanalysis. Our correlation analysis between meteorological factors and TAOD suggests the interannual variability of TAOD is related to the variability of precipitation (PPT), volumetric soil moisture (VSM), and wind speed (WS) in the particular regions. For instance, the positive TAOD trend over the equatorial central Pacific is mainly attributable to the increased PPT and decreased WS. In

contrast, in dry convective regions dominated by dust and smoke, the interannual variability/trend in TAOD is largely modified





by the VSM driven by the PPT. Additionally, we further found these significant regional correlations are more robust within the PBL and significantly weakened or even reversed within the FT. This highlights the superiority of using the TAOD partitioned within the PBL as a proxy variable for the widely applied TAOD to explore the relationships between atmospheric pollution and meteorology.

**1. Introduction**

Atmospheric aerosols (AAs), as an important constituent of the atmosphere, represent a key perturbation in the Earth's climate system. Their sources can be divided into natural sources (mainly from the breaking of air bubbles at the sea surface, wind erosion of soils, pollen bioaerosols, volcanic eruptions and forest wildfires), which are controlled by multiple factors such as the type of underlying surface, local meteorological factors, large-scale circulation patterns and climate forcing (De

Leeuw et al., 2011; Huneeus et al., 2011; Ginoux et al., 2012; Randerson et al., 2012), and anthropogenic sources (mainly from the burning of fossil fuels, industrial and agricultural activities and the gas–particle transformation of gaseous pollutants under certain conditions), which are influenced by the intensity of emissions from anthropogenic activities (Wang et al., 2014; Huang et al., 2015; Che et al., 2019; Zhang et al., 2019). Thus, different underlying surface characteristics, as well as regional differences in population density, industrial and economic development levels and meteorological conditions, together lead to

regional differences in aerosol concentrations, types and optical and microphysical properties to varying degrees.

In addition to adverse impacts on human health (Lelieveld et al., 2015), AAs can also influence the Earth's climate system in multiple and complex ways via aerosol–radiation interaction and aerosol–cloud interaction mechanisms. Generally, aerosol–radiation interaction alters the vertical thermal structure of the atmosphere via the effects of the scattering or absorption properties of AAs on solar and terrestrial radiation, which ultimately modifies the radiative balance of the Earth (Li et al., 2016;

Wilcox et al., 2016), leading to direct climate effects. In contrast, aerosol–cloud interaction has an indirect climate effect by altering the microphysical and radiative properties of clouds and their life cycles through the action of aerosols serving as cloud condensation nuclei or ice nuclei, thereby modifying precipitation (PPT) efficiency and atmospheric circulation patterns, which ultimately affect the global water cycle (Koren et al., 2012; Fan et al., 2015; Sarangi et al., 2018; Rosenfeld et al., 2019). Overall, the effect of aerosols on radiation, cloud, and PPT depends not only on the aerosol concentration, but also, and in

particular, on the types of aerosol (Jiang et al., 2018), their particle size distribution (Zhao et al., 2018), morphology (Zhao et al., 2020), chemical composition, and absorption properties (Ding et al., 2016; Ramanathan and Carmichael, 2008), as well as the environmental conditions.

On the other hand, besides the above-mentioned aerosol properties, it is also important to understand the vertical distribution (i.e., aerosol variability as a function of height) of different types of aerosols, particularly within the planetary

boundary layer (PBL). AAs have a short life cycle (about a week) owing to the effective scavenging of dry and wet deposition processes in the PBL (Bourgeois et al., 2018). Thus, the majority of the aerosol mass is considered to be present in the PBL, while the aerosol optical depth (AOD), i.e., the vertical integration of the aerosol extinction profile, is expected to be predominated by the contribution of the PBL. The dominant role of PBL aerosols has also been confirmed by previous studies,



which have found that the accumulation of PBL aerosols has a significant impact on weather, climate and air quality from
global to regional scales (Zhong et al., 2019; Huang et al., 2020). Specifically, the mass concentration of aerosols within the
PBL constrained by the vertical distribution of aerosols can influence the vertical profile of heating rates, which subsequently
changes the atmospheric stability and convection strength, potentially inducing more far-reaching effects on cloud and PPT
properties (Yu et al., 2019). Meanwhile, the indirect effects of aerosols depend mainly on those aerosols that are mixed with
(or are in contact with) clouds (Zhao et al., 2019), so it is essential to accurately characterize the three-dimensional (3D)
distribution of AAs. Furthermore, the health impacts of AAs are mainly related to the concentration of near-surface aerosols
that are inhaled into the human body (Zhao et al., 2018). More importantly, when aerosols (e.g., strongly absorbing smoke
aerosols) enter the free troposphere (FT) via convective transport, they may have a longer-term effect on climate because they
have a longer residence time (several weeks) than those in the PBL (Bourgeois et al., 2018; Yu et al., 2019). Therefore, besides
the column AOD, systematically and accurately characterizing the vertical profile of type-dependent aerosols (in particular
those partitioned in the PBL and FT), and their regional differences, as well as their interannual variation, is essential to
deepening our understanding regarding the role of aerosols in the Earth's climate system and human health.

The unavailability of accurate description on the vertical distribution of AAs is one of the main underlying factors
contributing to the large uncertainties that atmospheric models typically encounter when assessing aerosol direct radiative
forcing (Huneeus et al., 2011; Tian et al., 2017). In general, ground-based Lidar is considered to be the most effective way to
obtain the vertical distribution of aerosols. However, the limited spatial and/or temporal coverage of ground-based Lidar means
it can only be applied at small spatial scales, such as in a specific city or region. In September 2016, the launch of the Cloud–
Aerosol Lidar with Orthogonal Polarization (CALIOP) sensor onboard the Cloud–Aerosol Lidar and Infrared Pathfinder
Satellite Observation (CALIPSO) satellite enabled the detection of aerosol vertical profiles with unprecedented vertical
resolution and provided the first description of the 3D distribution of aerosols at a near-global scale. Since then, CALIOP data
have been extensively used in a variety of studies including, but not limited to: (1) describing the vertical distribution of
aerosols from global to regional scales (Adams et al., 2012; Tian et al., 2017; Zhao et al., 2018; Xu et al., 2018); (2) evaluating
model simulation outputs or reanalysis products in conjunction with other ground-based observations and remote sensing
instruments (Chen et al., 2018; Yao et al., 2020); and (3) assimilating into model simulations to correct for inhomogeneous
background error statistics (Sekiyama et al., 2010). Although these efforts have significantly improved our understanding of
the vertical distribution of aerosols and their climatic effects, a comprehensive study of the 3D climatology and spatiotemporal
variations of global and regional tropospheric type-dependent aerosols and their association with meteorological factors, based
on long-term (>10 years) continuous observations, is still lacking.

In this study, we utilized the last 13 years (2007–2019) of the Level-3 gridded tropospheric aerosol profile product, Version
4.2 (V4.2), retrieved from CALIOP to provide a synthetic description of the global climatology of tropospheric aerosols, with
particular attention paid to three climate-related aerosol types: dust, polluted dust (PD), and smoke. We also focus specifically
on the long-term evolution of the partitioning of the AODs for total aerosols and type-dependent aerosols in the total column
and specific layers, as well as those within the PBL and FT, and its association with interannual variations in meteorological



factors. Layered statistical sample data in CALIOP Level-3 products were also used to yield a 3D climatological distribution of the frequency-of-occurrence (FoO) of CALIOP-classified aerosol subtypes on a near-global scale.

## 2. Data and methods

### 2.1. CALIOP gridded tropospheric aerosol profile products

CALIOP is a dual-wavelength (532 and 1064 nm) polarization lidar onboard the CALIPSO satellite, launched on April 28, 2006. CALIPSO is a part of NASA's A-Train constellation of sun-synchronous-orbit satellites, with an orbital altitude of 705 km and a repetition period of 16 days. Currently, CALIOP is the only space-based satellite capable of providing continuous observations over many years, combining an active lidar instrument with visible imagers and a passive infrared sensor to observe the 3D vertical structure and optical properties of aerosols and clouds at high resolution on a near-global scale (82ºS–82ºN) (Winker et al., 2009, 2010, 2013). Global and regional observations obtained by CALIOP have been widely used to explore the role of AAs and clouds in regulating the Earth's climate system, and their interactions (e.g., Yu et al., 2017; Jiang et al., 2018; Sarangi et al., 2020).

In this study, we adopt the latest release of the Level-3 tropospheric aerosol profile product (Tropospheric_APro, V4.20) for "all-sky" conditions during both daytime and nighttime from 2007 to 2019, which is aggregated from the quality-screened Level-2 aerosol data (05kmApro, V4.20), including aerosol extinction profiles at 532 nm and layer classification information [i.e., aerosol, cloud, and "clear-air" (meaning air without aerosols)]. The CALIOP Level-3 aerosol profile product is based on monthly statistics, where all Level-2 aerosol profiles within each month are reported separately at a near-global scale (180°W to 180°E, 85°N to 85°S) on a uniform 2º × 5º (latitude × longitude) grid box with a vertical resolution of 60 m from −0.5 km to 12 km (total of 208 layers) above mean sea level (all altitudes hereafter are also above mean sea level). The reason for selecting 12 km as an upper limit in the Level-3 products is that the tropospheric aerosol content above 12 km in the Level-2 products is approximately 0.04%, which is negligible (Tackett et al., 2018). Compared to previous versions (V3.10), the V4.20 Level-3 aerosol profile product takes a series of more reasonable and strict quality filters to minimize the influence of Level-2 retrieval uncertainties, including extinction retrieval errors, layer classification errors, layer detection errors, and biases due to anomalous surface signal interruptions (Tackett et al., 2018). Here, we briefly summarize the quality filtering procedures used in generating Level-3 products, including (1) −100 ≤ cloud aerosol discrimination score ≤ −20; (2) extinction quality control flag = 0, 1, 16, or 18; (3) extinction uncertainty < 99.9 km$^{-1}$; (4) isolated 80-km aerosol layers are excluded; (5) aerosols in contact with ice clouds above 4 km are excluded; (6) aerosol extinction for "clear-air" = 0 km$^{-1}$, but the "clear-air" below aerosol layers with bases < 250 m is ignored; (7) extinction values below 60 m altitude (above ground level) are excluded; and (8) aerosol extinction with Cloud Fraction > 28 is rejected (Tackett et al., 2018).

In addition to providing aerosol vertical extinction profiles, CALIOP classifies aerosols into one of the seven aerosol subtypes—clean marine (CM), dust, polluted continental/smoke, clean continental (CC), PD, elevated smoke, and dusty marine (DM)—based on different physical characteristics, including surface type, geographical information (i.e., location and altitude),





particulate depolarization ratio, and integrated attenuated backscatter at 532 nm (Omar et al., 2009; Kim et al., 2018). Here, for the sake of simplicity, we refer to the 'polluted continental/smoke' as 'polluted continental (PC)' and the 'elevated smoke' as 'smoke' in this study. Each aerosol subtype is associated with an aerosol lidar ratio (i.e., extinction-to-backscatter ratio) determined from observations, modeling, and the cluster analysis of a multi-year AERONET (Aerosol Robotic Network) dataset, which allows the extinction of aerosols to be calculated from integrated attenuated backscatter (Omar et al., 2005,

2009; Winker et al., 2009; Young and Vaughan, 2009; Kim et al., 2018). Previous studies have demonstrated the reliability of CALIOP measurements, including aerosol extinction profiles and AOD, by comparing them with other observational techniques, such as ground-based lidar, sun photometers, and MODIS (Moderate resolution Imaging Spectroradiometer) observations (Schuster et al., 2012; Toth et al., 2016; Kim et al., 2018; Young et al., 2018). Furthermore, the classification of aerosol subtypes by CALIOP has been confirmed to be highly consistent with that of AERONET, especially for dust aerosols

(Mielonen et al., 2009; Burton et al., 2013). Therefore, in this study, in addition to focusing on the vertical extinction profiles and AOD at 532 nm for all aerosol types, special attention is paid to the following three individual aerosol subtypes: dust, PD, and smoke. Meanwhile, the sample statistics for all seven aerosol subtypes are also considered in this study. Additionally, CALIOP has reportedly been experiencing low-energy laser shots since September 2016, primarily within and near the South Atlantic anomaly region (see https://www-calipso.larc.nasa.gov/resources/calipso_users_guide/advisory.php), which severely

affects the scientific quality of the profile observations in the region. Therefore, to avoid this anomaly affecting the quality of Level-3 data, all Level 2 profiles for post-2017 data within the South Atlantic anomaly region were excluded when constructing the Level-3 product. Fig. S1 shows the excluded region for the period 2017–2019. As a result, all analyses in this paper within the South Atlantic anomaly region are based on the period 2007–2016.

Note that although solar background illumination during daytime reduces the signal-to-noise ratio, which leads to a better

aerosol extinction detection sensitivity during the nighttime, we do not use the nighttime data in isolation for the analysis here; instead, the average values of daytime and nighttime data (including extinction profiles and sample statistics for all aerosol types and different aerosol subtypes) are utilized. On the one hand, this is because the same quality screening is used for Level-3 products during both daytime and nighttime. On the other hand, by averaging daytime and nighttime data, the effect of diurnal variability in aerosol loadings is reduced.

In order to explore the climatological distribution characteristics and evolutionary trends of type-dependent aerosols and their FoO with altitude, this study divides the 0–12-km troposphere into six specified altitude layers (0–2 km, 2–4 km, 4–6 km, 6–8 km, 8–10 km, and 10–12 km) at 2-km intervals from the surface to 12 km. For the extinction profiles due to all aerosol types and type-dependent aerosols (dust, PD, and smoke), we calculated the AOD at a specified altitude layer ( $\text{AOD}_{\text{layer-specific}}$ ) from the vertical extinction coefficient at 532 nm as follows:

$$\text{AOD}_{\text{layer-specific}} = \int_{z_1}^{z_2} \text{AEC}(z)dz \, , \tag{1}$$



where AEC($z$) represents the aerosol extinction coefficient for all aerosol types and type-dependent aerosols at altitude $z$, and $z_1$ and $z_2$ are the minimum and maximum altitude of the specified altitude layer, respectively. The ratio (%) of $AOD_{layer\text{-}specific}$ to total column AOD ( $AOD_{total}$ ) in the troposphere was also evaluated, according to the following equation:

$$R_{layer\text{-}specific} = \frac{AOD_{layer\text{-}specific}}{AOD_{total}} \times 100.0\% \, , \tag{2}$$

Moreover, in addition to exploring the stratified AOD and its partitioning, this study also pays particular attention to the 3D distribution characteristics of FoO with altitude for the CALIOP-derived aerosol subtypes. Consistent with previous studies (Adams et al., 2012; Liu et al., 2019), the FoO (%) of type-dependent aerosols at different altitudes is defined as

$$FoO = \frac{N_{aerosol}}{N_{total}} \times 100.0\% \, , \tag{3}$$

where $N_{aerosol}$ represents the number of samples detected by CALIOP for a specific aerosol type at the specified altitude layer, 
and $N_{total}$ is the total number of samples (including all aerosol types and "clean air") at the specified altitude layer. At the same time, the total column (i.e., all layers) FoO (%) for each aerosol subtype at each grid box was further calculated according to the following equation:

$$FoO_{all\ layers} = \frac{\sum_{i=1}^{i=208} N_{aerosol}^{i}}{\sum_{i=1}^{i=208} N_{total}^{i}} \times 100.0\% \, , \tag{4}$$

where $N_{aerosol}^{i}$ represents the number of samples for a specific aerosol type at the specified altitude layer $i$ (maximum number 
of layers is 208), and $N_{total}^{i}$ is the total number of samples (including all aerosol types and "clean air") at the specified altitude layer $i$.

## 2.2. Multi-sensor and reanalysis AOD datasets

In order to ensure the accuracy of the AOD trends assessed by CALIOP, five different datasets of AOD at 550 nm, obtained from three satellite retrievals and two aerosol reanalyses, are also used for intercomparison purposes. The three 
satellite-based AOD retrieval datasets are from MODIS (onboard the Terra and Aqua satellites) and the Multi-angle Imaging Spectro-Radiometer (MISR) (onboard the Terra satellite). In this study, we use the monthly gridded AOD product Collection C6.1 (MYD08_M3 and MOD08_M3 for the Aqua and Terra satellites, respectively), with a 1° × 1° resolution, covering the period 2007–2019, derived from the combined Dark Target and Deep Blue algorithms. MISR monthly AOD data with a 0.5° ×0.5° resolution were obtained from the Level-3 global aerosol product (MIL3DAE), version F15_0032.

The two AOD reanalysis datasets are from the Modern-Era Retrospective Analysis for Research and Applications, version 2 (MERRA-2) and the Copernicus Atmosphere Monitoring Service (CAMS). MERRA-2 is a second-generation atmospheric aerosol reanalysis, which assimilates both meteorological observations and the AODs from various ground- and satellite-based



observations (Buchard et al., 2017). CAMS is the latest global atmospheric composition reanalysis dataset generated by the European Centre for Medium-Range Weather Forecasts and consists of atmospheric composition fields including aerosols,
major precursors, and chemical species (Inness et al., 2019). Monthly mean AOD reanalysis data from MERRA-2 (0.5° latitude by 0.625° longitude) and CAMS (0.75°by 0.75°) are used in this study.

### 2.3. Soil moisture and PPT data

To explore the interannual variations in the total AOD (TAOD), PBL and FT AODs for smoke and dust aerosols and their linkages with major meteorological controlling factors such as surface wind speed, volumetric soil moisture (VSM), and
PPT, monthly mean data of wind speed (WS) at 10 m from the MERRA-2 reanalysis, VSM at a depth of 0–10.0 cm from the Global Land Data Assimilation System (GLDAS), and PPT from the Global Precipitation Climatology Project (GPCP), version 2.3 (Adler et al., 2003) are used.

GLDAS characterizes the state of the global land surface condition by integrating multi-satellite and ground-based datasets and using data assimilation techniques and land surface modeling (Rodell et al., 2004). GLDAS/NOAH10 (V2.1)
VSM monthly mean products at a spatial resolution of 1.0° × 1.0° are available for 2000 to the present day. We chose this product because of its relatively high spatial resolution and because it is widely used to verify soil moisture content in satellite retrievals (Dorigo et al., 2010). GPCP, from the National Oceanic and Atmospheric Administration, is a global combined PPT dataset available monthly from 1979 to the present day at a 2.5° × 2.5° resolution. This dataset is derived from multi-source satellite data combined with land-based rain gauges and sounding observations (Adler et al., 2003). Its long time scales and
global coverage make it suitable for studying the connections between type-dependent aerosols and PPT. In this study, the monthly mean WS, VSM and PPT data were bilinearly interpolated to a spatial resolution of 2.0° × 5.0° to match the resolution of the CALIOP Level-3 products.

### 2.4. Extraction of PBL AOD from column AOD

Partitioning of the AODs for all aerosol types and type-dependent aerosols in the PBL and FT (the PBL top to 12 km)
was computed from the gridded CALIOP Level-3 aerosol extinction profile and the PBL height (PBLH) product from the MERRA-2 reanalysis. In this study, the hourly (00:30, 01: 30, …, 23:30 UTC) MERRA-2 PBLH data during 2007–2019, with a spatial resolution of 0.5° × 0.625°, were used. The MERRA-2 PBLH is diagnosed from a turbulence parameterization module based on the thermal eddy diffusion coefficient in the general atmospheric circulation model (Jordan et al., 2010). Compared to the PBLHs obtained by other observation techniques (e.g., retrievals from micropulse lidar, radiosondes, and CALIOP),
several previous analyses have demonstrated that MERRA-2 is sufficiently robust to capture the daily, monthly, seasonal, and annual variations in regional PBLHs (Jordan et al., 2010; Su et al., 2017, 2018).

Since CALIPSO is polar sun-synchronous satellite with fixed equatorial crossing times [01:30 and 13:30 local time (LT)], the hour-by-hour MERRA-2 PBLH data at UTC needed to be converted to LT corresponding to daytime (13:30 LT) and nighttime (01:30 LT) CALIOP observations, respectively. Specifically, for each hourly field of PBLH data at a given UTC,



the associated LT was computed by adding the UTC of 1 hour every 15° longitude away from the prime meridian in the easterly direction. Then, we extracted the hourly PBLH fields corresponding to 01:30 and 13:30 LT, respectively, for further monthly averaging. To complete the spatial analysis, the monthly averaged MERRA-2 PBLH data for the two LTs were bilinearly interpolated to the spatial resolution of the Level-3 aerosol product from CALIOP ($2º \times 5º$). Next, we averaged the monthly PBLH data at 01:30 and 13:30 LT, and then assigned them to the monthly mean extinction profiles (average value of daytime

and nighttime) by using the corresponding CALIOP Level-3 grid box. Finally, the monthly PBLH value determined for each grid box was then used to separate the monthly mean PBL and FT AOD by using Eq. (1).

**2.5. Statistical analysis**

To reveal the effects of meteorological factors on the interannual variability (or trends) in aerosol loadings for the period 2007–2019, we performed a Pearson's correlation analysis between smoke and dust aerosols and three dominant

meteorological factors: WS, VSM, and PPT. Long-term trend analysis of the total column AOD, layer-specific AOD, and PBL and FT AODs for all aerosol types and type-dependent aerosols were conducted on annual time series, using the Mann–Kendall (M-K; Mann, 1945; Kendall, 1957) test with Sen's slope estimation, a widely used technique for trend assessment (Achakulwisut et al., 2017; Li et al., 2018; Jin and Pryor, 2020; Gui et al., 2021). To ensure the robustness of the trend assessment, we needed at least 60% of the data in each annual time series to be valid before the trend calculations were

performed.

**3. Results and Discussion**

**3.1. 3D climatological distribution of total and type-dependent aerosols**

The multi-year (2007–2019) average of the total aerosol extinction coefficient (TAEC) is clearly depicted in Fig. 1 with an innovative global 3D view, which is much more intuitive than the traditional two-dimensional or slice-by-slice

presentation (see Fig. S2). The global 3D distribution pattern of total aerosols is the consequence of the interaction between aerosol emissions from natural and anthropogenic sources and meteorological drivers (Winker et al., 2013). Regional differences in aerosol concentration, type and vertical distribution may be one of the main reasons why the extent of aerosol effects on radiation, clouds and PPT varies significantly between regions (Jiang et al., 2018). Overall, the high values (0.05–0.1 $km^{-1}$) of TAEC over the oceans are mostly concentrated below 1 km. The main contributors to this TAEC enhancement in

the lower layers over the oceans are wind-driven sea spray particles, which are composed of sea salt aerosols and marine primary organic matter. In contrast, the vertical distribution of TAECs over land has distinct regional differences, with high values (> 0.1 $km^{-1}$) generally located in areas such as source areas of natural mineral dust and high anthropogenic emissions, and its magnitude tends to decrease with increasing altitude. Affected by multiple factors such as emission intensity, meteorological conditions, and terrain uplift, aerosols can be elevated above 8 km. This phenomenon is prevalent in several

regions of the world, such as a high-altitude, high-concentration aerosol belt that is clearly visible from 10° to 40 °N latitude.



A similar, but much less intense, aerosol belt is apparent in the area north of 40°N latitude. The formation of this aerosol belt is primarily associated with the high-altitude transport of dust and smoke aerosols.

The differences in the optical properties (i.e., absorption and morphology) of different aerosol types allow the CALIOP retrieval algorithm to assign the total aerosol extinction to aerosol subtypes. Fig. 2 presents the 3D distribution of AEC for the three main aerosol types (dust, PD and smoke) classified by CALIOP. These three aerosol species were chosen because of their remarkable climatic effects. In the CALIOP V4.2 classification algorithm, the "dust" type represents uncontaminated pure dust, the "PD" type tends to represent a mixture of dust and PC aerosols or smoke aerosols, and the "smoke" type represents the lifted smoke plumes produced by biomass-burning emissions, which consist mainly of carbonaceous aerosols with strong absorption of black carbon and organic carbon. For dust aerosols, the enhancement of dust AEC (DAEC) is located mainly in arid and semi-arid regions, including the Sahara and Sahel, the Arabian Peninsula, northern India, the Tarim Basin and the Gobi Desert, which are the main contributors of dust globally, and form the well-known dust belt in the Northern Hemisphere. Vertically, dust aerosols can be lifted up to an altitude of 6 km, and with their high values usually located in the near-surface layer (below 3 km) nearby the dust source. In addition, this 3D structure provides direct observational evidence for the widely identified dust trans-Pacific transport phenomenon. That is, mineral dust from desert source areas in East Asia, driven by westerly or northwesterly winds, can be transported all the way along the North Pacific Ocean to near the United States West Coast, with a maximum uplift altitude of 4–6 km (also see Fig. S2b). Similarly, consistent with previous findings (e.g., Adams et al., 2012; Yu et al., 2015), the trans-Atlantic transport phenomenon of dust aerosols is also confirmed. The results show that dust aerosols from the Sahara Desert (SD) can be transported to North and Central America, thus forming a dust belt that can extend to the Caribbean and northern South America, with a maximum uplift altitude of 6–7 km.

Unlike the 3D pattern of dust aerosols, PD is more widely distributed—mostly in regions downstream of the dust source where there are intensive human activities or typical biomass burning, e.g., the eastern United States (EUS), Amazon Zone (AMZ), eastern Africa, southern Asia (SA), and northwestern and northern China. Enhanced PDAEC is usually constrained to altitudes below 3 km near the ground, which can be mainly attributed to the distribution related to anthropogenic pollutant emissions. For smoke aerosols, the distribution of smoke AEC (SAEC) has a distinct regional dependence. Specifically, its high values are mainly distributed in typical biomass burning regions, including central-southern Africa (CSA), the AMZ, southeastern Asia (SEA), and central Siberia, where the maximum lifting altitude of smoke aerosols exceeds 4–6 km. Given their strong absorption properties, the wide distribution range of smoke aerosols has been proven to have a non-negligible impact on weather and climate. For instance, previous studies have shown that smoke plumes generated by forest fire emissions can even be ejected into the stratosphere, where they can remain for up to eight months (Yu et al., 2019). In addition, black carbon aerosols—one of the main components of smoke plumes—entering the stratosphere, can also affect the dynamic stability and horizontal circulation of the stratosphere by heating the surrounding air, thus disturbing the radiative balance of the Earth system and ultimately predisposing it to drastic global climate change.

The climatological distributions of the whole-layer integrated AODs for total aerosols and for aerosol sub-types are given in Fig. S3. The CALIOP-based observations show that the global average AOD of total aerosols reaches 0.095, of which dust,




PD and smoke aerosols account for 0.020, 0.011 and 0.008, respectively. In terms of type-dependent AODs as a percentage of TAOD, dust AOD (DAOD), polluted dust AOD (PDAOD) and smoke AOD (SAOD) contribute 15.55%, 10.01% and 7.81% of TAOD, respectively (Fig. S4), while the remaining 66.63% is contributed by other types of aerosol (OTA), including anthropogenic pollution aerosols consisting of sulfate, nitrate and ammonium, and sea-salt aerosols. Spatially, TAOD is similar to the distribution patterns previously obtained based on ground-based observations, satellite retrievals, and model simulation

(Hsu et al., 2012; Chin et al., 2014; Che et al., 2015). In contrast, there are significant regional differences in the spatial distribution of type-dependent AODs, which can mainly be attributed to the regional differences in anthropogenic activity intensity, surface type, and climatic conditions.

       Fig. 3 further presents the distribution characteristics of the layer-specific integrated AODs (at 2-km intervals) with altitude for these aerosol species. The results show that the layer-specific integrated TAOD markedly decreases with altitude,

and about 80% or more of the aerosols are located in the lower troposphere (0–2 km altitude range) (Fig. S5). Aerosols over land can still contribute about 10% of the column aerosol loading in the altitude range of 2–4 km, which is indicative of the complexity of aerosol sources over land and their interactions with the topography and meteorological conditions. Compared with aerosols over land, the aerosol loading over the oceans shows a rapid decline with altitude, and the percentage of aerosol loading within 2–4 km drops to less than 5% in all regions except for the ocean areas adjacent to land. When the altitude

exceeds 6 km, the high values of TAOD are mainly located in the middle and high latitudes of the Northern Hemisphere, and the main contributors are elevated dust and smoke aerosols. When the TAOD is separated according to aerosol types, differences in the distribution of layer-specific integrated AOD with altitude are apparent for different types of aerosols. Specifically, dust aerosols are scattered almost equally in the altitude ranges of 0–2 km and 2–4 km, and their enhancement is mainly located in source areas and their downstream. Vertically, the elevated altitude range of dust aerosols can extend from

near the ground all the way to more than 8 km, which is mainly the result of the interaction between dust, terrain and wind (Xu et al., 2018).The latest CALIOP aerosol type discrimination algorithm further classifies the mixed dust aerosols over the oceans into PD and DM. Thus, between 0–2 km, PD is almost entirely distributed over land, and the high values of its proportion (> 60%) are mainly distributed in continental regions north of 40°N, northern Africa, coastal regions of South America, northern SA, and eastern China. In contrast, the proportion of PD is higher over the oceans than over land in the altitude range above 4

km. Unlike dust and PD aerosols, smoke aerosols are slightly more abundant in 2−4 km than in 0−2 km, and the vertical impact extent of smoke aerosols can be extended all the way to about 10 km. Overall, the proportion of smoke aerosols on land decreases rapidly with altitude, while the proportion on the ocean does not show a monotonic decreasing trend.

**3.2. Partitioning of TAOD and type-dependent AODs between the PBL and FT**

       Accurate partitioning of the loadings in total aerosols and key aerosol types within the PBL and FT is essential for

constraining atmospheric models and accurately quantifying the radiative effects of aerosols. Here, the partitioning of the AOD in the PBL and FT was calculated using the monthly CALIOP Level-3 aerosol extinction retrievals and the PBLH data from the MERRA-2 Reanalysis (see section 2). The annually averaged global distribution of PBLH and FT AODs for all aerosol





types, dust, PD and smoke over 13 years (2007–2019) are shown in Fig. 4. Overall, for all aerosol types, the global TAOD is 0.0923 in the troposphere, of which 0.0571 is found in the PBL and 0.0352 in the FT, corresponding to 61.86% and 38.13%, respectively, of the TAOD. The above results are largely consistent in terms of the average partitioning between daytime and nighttime (daytime: 69% and 31% within the PBL and FT, respectively; nighttime: 38% and 62% within the PBL and FT, respectively) obtained by Bourgeois et al. (2018) using Level-2 CALIOP aerosol extinction retrievals during 2007–2015. The slight difference between the two can be mainly attributed to the differences in the PBLH products used and the time-matching scheme. However, if the low bias of the CALIOP aerosol extinction in the lower troposphere (below 2 km) with respect to surface lidar observations noted in prior studies (e.g., Campbell et al., 2012; Papagiannopoulos et al., 2016) is taken into account, the contribution of column TAOD within the PBL determined in this study is expected to be even higher than 62%. Spatially, high values of TAOD can be seen both in the PBL and FT over west-central Africa, the Arabian Peninsula, India and eastern China. This distribution is expected since these regions correspond to the main continental sources of aerosol mass. Typical features such as aerosol transport over the Arabian Sea and the Bay of Bengal or long-range transport of dust aerosols from the Sahara over the Atlantic Ocean are also observed in the FT assigned by CALIOP data.

Unlike TAOD, which accounts for approximately 1.6 times more in the PBL than in the FT, type-dependent aerosols account for significantly different proportions in the PBL and FT. For example, although the locations of high values of DAOD and PDAOD over land are very different, the AOD partitioning between the PBL and FT is similar for dust (49.48% and 50.51%, respectively) and PD (52.22% and 47.79%, respectively). In contrast, a very different partitioning ratio (23.68% and 76.32% for the PBL and FT, respectively) is apparent for smoke aerosols. For dust aerosols, the high DAOD within the PBL is mainly located in the dust source area, whereas its high values within the FT are located in the downstream region away from the dust source area. The enhanced DAOD within the FT contributes mainly to two regional processes: pyroconvection and orographic lifting, which can transport dust aerosols from the surface to the FT (Yumimoto et al., 2009; Bourgeois et al., 2015). For PD aerosols, the contribution of PDAOD is more or less equal in the PBL and FT over most land areas, except in SA and eastern China where the PDAOD within the PBL is higher than that within the FT. The equivalent partitioning of PDAOD within the PBL and FT is mainly attributable to the adequate mixing of dust aerosols and anthropogenic pollution aerosols within the PBL, as well as to pyroconvection and orographic lifting. For smoke aerosols, the partitioning of SAOD within the FT is almost three times as much as that within the PBL, indicating that light-weight, fine-mode carbonaceous aerosols are more likely to be lifted to higher altitudes by horizontal and vertical motions with a very wide range of effects. Overall, the degree of influence of smoke aerosols within the PBL is much weaker than that within the FT. This is evidenced by the fact that biomass-burning aerosols released from the China-Indochina Peninsula, CSA and AMZ produce a broader range of effects within the FT than within the PBL.

### 3.3. Regional-averaged vertical distribution of TAEC and type-dependent AECs with altitude

Following our previous studies (Gui et al., 2021), we selected 12 regions of interest (ROIs) that have suffered from enhanced anthropogenic or natural aerosol pollution and have received extensive attention in other aerosol-related studies: the





EUS, Western Europe (WEU), northwestern China (NWC), northeastern Asia (NEA), northern China (NC), southern China (SC), the SD, Middle East (ME), AMZ, CSA, SA, and SEA. Detailed geographic information for these 12 study regions is presented in Table S1. These ROIs are used to explore the regional differences in the vertical extinction profiles and FoOs for different types of aerosols. Fig. 5 presents the regional-averaged vertical profiles of the multi-year average TAEC over the 12

ROIs. The vertical profile of the multi-year averaged TAEC is somewhat representative of the climatological characteristics of the vertical distribution pattern of total aerosol concentration. Comparing the vertical profiles of TAEC for different ROIs shows that the peaks of regional TAEC occur, in descending order, in NC ($0.47\ \text{km}^{-1}$), ME ($0.37\ \text{km}^{-1}$), SA ($0.32\ \text{km}^{-1}$), SC ($0.31\ \text{km}^{-1}$), SD ($0.18\ \text{km}^{-1}$), AMZ ($0.18\ \text{km}^{-1}$), NWC ($0.16\ \text{km}^{-1}$), CSA ($0.14\ \text{km}^{-1}$), WEU ($0.12\ \text{km}^{-1}$), SEA ($0.10\ \text{km}^{-1}$), NEA ($0.08\ \text{km}^{-1}$), and EUS ($0.07\ \text{km}^{-1}$). The vertical profile of TAEC is dominated by a single peak in most regions, and the

altitude corresponding to the peak is mostly below 1 km. Among them, TAEC reaches a maximum near the ground in NC and SA, and then shows a rapid decreasing trend with altitude. In CSA and NWC, it is multi-peaked, with the former having a smaller peak around 3 km, mainly attributable to the uplifted smoke aerosols. In addition, the vertical profile of TAEC in NC is accompanied by large interannual fluctuations (corresponding to amplified standard deviations) below 3 km. This large interannual variation in TAEC is mainly attributable to the decreased anthropogenic emissions as a consequence of clean-air

actions implemented by the Chinese government in the last decade (Zheng et al., 2018; Gui et al., 2019), resulting in a rapid decline in annual aerosol concentrations from the near-surface to upper layers (see Fig. S6). Similar interannual evolutionary patterns of the vertical profile of TAEC are also apparent for SC, EUS and WEU.

Fig. 6 presents the climatology of the regional-average profiles over the 12 ROIs for the three aerosol types (dust, PD and smoke). Similarly, the year-to-year variation of these aerosol types is presented in Figs. S7–S9. Influenced by the mixing of

anthropogenic aerosols and dust aerosols, PDAEC is dominant in the EUS, WEU, SA, NC and SC, and the maximum values of PDAEC are distributed below 2 km. Notably, in the EUS, WEU, NC and SC regions, the annual PDAEC profiles show a leftward shift over time, indicating a decreasing trend in PDAEC over the last 13 years (Fig. S8), which is consistent with the trends in anthropogenic emissions over the above regions revealed in previous studies (Zhao et al., 2017). In contrast, the SA region shows a clear rightward shift, corresponding to the increasing anthropogenic emissions in the region (Che et al., 2019).

However, a similar evolutionary trend (rightward/leftward shift) is not apparent in other ROIs. The enhanced DAEC profile occurs mainly in dust source regions, including the ME, SD and NWC. In the ME, the DAEC reaches a maximum ($\sim 0.30\ \text{km}^{-1}$) near the ground and then decreases gradually with altitude. In SD, it reaches a maximum ($\sim 0.10\ \text{km}^{-1}$) at an altitude of 0.5 km. In contrast, the DAEC peak in NWC occurs at an altitude of 1.5 km. Compared to the PDAEC profile, a more pronounced interannual fluctuation is found in the DAEC from the lower to the middle and upper levels (Fig. S7). In NEA, AMZ, CSA

and SEA, the vertical aerosol extinction is dominated by elevated smoke, and the vertical distribution of SAEC shows a bimodal distribution, with the peak locations at 0–1 km and 2–4 km, respectively. Although more prominent interannual differences are apparent in the AMZ and SEA regions than other ROIs, the altitudes corresponding to the peak of the SAEC profile do not change significantly (Fig. S9). It is also noteworthy that the elevated smoke aerosol layer is prevalent over the anthropogenic aerosol–dominated EUS, WEU and SA too, with the peak SAEC corresponding to altitudes of about 2–4 km.





Fig. 7 presents the statistical results of the multi-year average of the integrated AOD of TAEC over the globe, land, ocean, and 12 ROIs within a specific layer at 2-km intervals. Meanwhile, the statistics of the relative proportion (%) of the layer-specific AODs to TAODs are also shown in Fig. 7. The results show that, on a multi-year global average, the contribution of the integrated AOD within 0–2 km, 2–4 km, 4–6 km, 6–8 km, 8–10 km, and 10–12 km to TAOD is 80.38%, 15.41%, 3.42%, 0.52%, 0.20% and 0.07%, respectively. In contrast, the contribution of the integrated AOD within these five specific altitude

ranges is 70.82 (87.52%), 23.59% (9.41%), 4.82% (2.28%), 0.53% (0.51%), 0.18% (0.21%) and 0.06% (0.07%) over land (ocean), respectively. Ocean–land differences in the contribution of the lower troposphere suggest that aerosols over land are capable of being lifted to higher altitudes by the topography and atmospheric circulation. In all 12 ROIs, more than 50% of the aerosol loading is located in the lower troposphere in the 0–2 km altitude range, while the contribution is less than 2% in altitudes above 6 km. Among these 12 ROIs, the proportion of integrated AOD within 0–2 km is located, in descending order,

in WEU (85.12%), SEA (83.54%), NC (81.68%), EUS (80.67%), SC (79.51%), NEA (77.44%), AMZ (77.08%), SA (75.50%), ME (65.98%), SD (58.66%), NWC (54.58%), and CSA (51.91%). The proportion of the layer-specific integrated AOD at 2–4 km to TAOD decreases more than two to three times compared to that at 0–2 km over all ROIs, except NWC and CSA. Among them, the highest value of regional aerosol loading within 2–4 km, as a percentage of total aerosol loading, occurs in NWC (36.37%), and the lowest occurs in WEU (12.12%). In contrast, the highest and lowest values at 4–6 km occur in the

ME (8.15%) and AMZ (1.84%), respectively. The percentage of layer-specific integrated AOD to regional TAOD is less than 1%, 0.4% and 0.2% for all ROIs, except NEA at 6–8 km, 8–10 km and 10–12 km, respectively.

Fig. 8a further quantifies the contribution of different types of aerosols in the multi-year regional average TAOD. Results show that the multi-year global average TAOD is 0.095, with dust, PD, smoke and OTA contributing 0.020, 0.011, 0.007 and 0.056, respectively. Compared to the global average, the multi-year land (ocean) averaged TAOD is higher (lower) at 0.117

(0.085), with dust, PD, smoke and OTA contributing 0.040 (0.010), 0.031 (0.002), 0.012 (0.005) and 0.033 (0.067), respectively. This pattern of higher TAOD over land than ocean does not vary across the altitude range (Fig. S10). However, the contribution of dust and smoke to the integrated TAOD within the specified altitude is dominant with increasing altitude, both over land and ocean.

Among the 12 ROIs, the multi-year regional-averaged TAOD values, from largest to smallest, occur in NC (0.427), ME

(0.415), SA (0.382), SC (0.353), SD (0.315), CSA (0.258), NWC (0.193), NEA (0.140), AMZ (0.132), SEA (0.124), WEU (0.104), and EUS (0.091). Of these, dust, PD, smoke and OTA contribute 0.067, 0.190, 0.034 and 0.136, respectively, to the largest TAOD in NC. In contrast, dust, PD, smoke and OTA contribute 0.008, 0.021, 0.013 and 0.049, respectively, to the lowest regional-averaged TAOD in the EUS. In terms of regional averages for type-dependent aerosols, we find that the maximum value (0.312) of the regional-averaged DAOD occurs in the ME, while the minimum value (0.007) occurs in CSA.

In contrast, the maximum value of the regional-averaged PDAOD (0.190) occurs in NC, while the minimum value (0.014) occurs in WEU. For SAOD, the maximum regional-average value (0.117) occurs in CSA, while the minimum value (0.002) occurs in the ME. From different altitude ranges, the distribution modalities of the regional-averaged whole-layer TAOD





remain largely consistent with the 0–2 km range (see Fig. S10). However, with increasing altitude, the TAOD values over different ROIs vary significantly, both in terms of magnitude and aerosol type contribution.

**3.4. Climatology of FoOs for total and type-dependent aerosols**

Based on layered statistical sample data of seven types of aerosols (i.e., CM, dust, PC, CC, PD, smoke, and DM) in the CALIOP Level 3 aerosol profile product, we calculated the FoOs for different types of aerosols at the whole-layer scale and at specific altitudinal locations globally and over the 12 ROIs, respectively. The global distribution of the whole-layer FoOs for all aerosol types and aerosol sub-types is shown in Fig. 9. As expected, the high values (~10%–20%) of cumulative FoOs
for all aerosol types (Fig. 9h) are similar to the TAOD (see Fig. S3a) in terms of spatial distribution patterns, and are located in areas of high aerosol emissions from anthropogenic or natural sources. Although a similar spatial distribution pattern of TAOD and FoO is observed, in contrast to the maximum values of TAOD, which are located in SA and eastern China, the maximum values of FoO (~15%–25%) are mostly located in North Africa, the ME, and their adjacent seas. Regionally, the second highest values of FoO (~8%–15%) are observed in SA, NC, eastern China, EUS, southern Africa, the tropical mid-
Atlantic (extending all the way to the Caribbean Sea in central North America), and the central Pacific.

By separating the whole layer of accumulated FoO for all aerosol types according to aerosol sub-types, the independent distribution pattern of total FoO contributed by multiple aerosol types is clearly presented (Figs. 9a–g). The results show that, among all CALIOP-discriminated aerosol types, dust and PD aerosols are the most frequent aerosol types over land globally, with the former occurring most frequently (~10%–15%) in desert source areas, while the occurrence frequency of the latter is
more widely distributed, with high values (~5%–10%) mainly located in desert source areas and adjacent downstream regions (e.g., SA, NC, and eastern Africa). The high values of whole-layer FoO for PC aerosols are mainly distributed in regions with enhanced anthropogenic aerosol emissions (closely linked to population density and industrial development levels), and their high values (~5%) are mainly distributed in regions such as eastern China, SA, CSA, WEU, EUS, northern Australia, and northern South America. In contrast, the FoO for CC aerosols is globally low and almost negligible. For smoke aerosols, the
enhanced whole-layer FoO (~3%–5%) is mainly distributed in CSA and adjacent seas, northern Australia, SC, the Indochina Peninsula, and central North America (mainly Mexico). DM aerosol is a newly categorized aerosol type in the latest version of CALIOP, which can represent the mixed state of dust and sea salt aerosols. The results show that the regions experiencing the most frequent mixing of dust and sea salt aerosols are mainly located in the tropical mid-Atlantic, Mediterranean Sea, Arabian Sea, Bay of Bengal, and East China Sea, where the whole-layer FoO can reach 5%–10%.

Similar to the TAOD, we also present the statistics of the multi-year regional-average FoO over the globe, land, ocean, and 12 ROIs (Fig. 8b). In general, a higher regional-average FoO corresponds to higher TAOD values. Specifically, on a global average, the multi-year averaged whole-layer FoO for all aerosol types is 4.45%, with CM, dust, PC, CC, PD, smoke and DM aerosols contributing 1.79, 0.86, 0.30, 0.05, 0.57, 0.38 and 0.50%, respectively. In contrast, the multi-year averaged FoO over land (ocean) is 4.64% (4.41%), with CM, dust, PC, CC, PD, smoke and DM aerosols contributing 0.10 (2.58), 1.76 (0.44),





0.62 (0.14), 0.15 (0.04), 1.44 (0.17), 0.48 (0.34) and 0.08% (0.70%). Among the 12 ROIs, the maximum value of region-averaged FoO occurs in the ME (16.43%), while the minimum value occurs in WEU (4.63%).

Fig. 10 presents the 3D distribution of FoO for all aerosol types at the specified 208 layers from the altitude range of 0 to 12 km. Also, the 3D distribution of FoO for the seven aerosol sub-types classified from CALIOP are shown in Fig. 11 and Fig. S11. On a global scale, the 3D distribution pattern of cumulative stratified FoO for all aerosol types with altitude remains
highly consistent with that of TAEC (Fig. 1). In land areas other than the desert source, high values of stratified FoO (generally > 60%) occur in the lower atmosphere (below 1 km). Vertically, stratified FoO shows a decreasing trend with altitude. In the dust source area, the high values of stratified FoO (70%) can extend from the surface to about 3 km. Over the ocean, near the surface is the altitude where sea salt aerosols are most frequently detected.

The 3D distribution of the FoO for dust aerosols (Fig. 11a) shows that the enhanced stratified FoO at different altitudes
is located in the dust belt spanning the Sahara, the ME and NWC, with the highest FoO (~60%) located at altitudes of 1–3 km. It is noteworthy that most regions of the Northern Hemisphere suffer from trans-regional transport of dust aerosols, which is continuously lifted by the topography, enabling the stratified FoO to reach 3%–5% within 4 km in most regions of the Northern Hemisphere except for the dust source. Meanwhile, the maximum value of the stratified FoO (~3%) of dust aerosols, originating from the desert source area in NWC during the prevailing trans-Pacific transport, is located at 3–4 km altitude.
Unlike dust aerosols, high values (~20%–30%) of the stratified FoO for PD aerosols are mainly located in the dust source area, as well as in the downwind region, with a spread altitude of about 0–4 km (Fig. 11b). Conversely, the maximum values of the stratified FoO of PC aerosols are all located at altitudes below 1 km and are mainly controlled by anthropogenic aerosol emissions near the ground (Fig. 11d). In contrast, the maximum value of the stratified FoO for smoke aerosols is located at 3 km altitude in CSA and its adjacent oceans (Fig. 11c). For DM aerosols, the maximum value of FoO (~30%) is located in the
altitude range of 0–2 km over the Arabian Sea between Saudi Arabia and SA (Fig. 11e). Also, high FoO (~10%–20%) is observed over the tropical mid-Atlantic (at an altitude of about 0–3 km). Further analysis of the 3D distribution of FoO for both clean-type aerosols (i.e., CC and CC) shows that, in most ocean areas far from the continents, CM aerosols are found to exhibit high FoO values (> 60%) in the lower atmosphere. Instead, in the lower atmosphere over the oceans close to the continents (0–2 km), the values are able to decrease to 10%–30% (Fig. 11f). Unlike CM aerosols, which exhibits high FoO
over most of the ocean, the stratified FoO for CC aerosols is low over most land areas (see Fig. S11).

### 3.5. Vertical profiles of regional-averaged FoOs for different types of aerosols

Fig. 12 presents the multi-year regional-averaged vertical profiles of FoOs for the seven types of aerosols over the 12 ROIs and their interannual fluctuations. Overall, the stratified FoO for different types of aerosols shows contrasting vertical distribution characteristics over the 12 ROIs. The results show that CC aerosols are less frequent in all ROIs compared with
other aerosol types, and the maximum regional-average stratified FoO occurs at an altitude of 0.5 km in the EUS with a value of 2.1%. In contrast, the maximum value of the stratified FoO over all other ROIs is less than 1.2%. CM aerosols are frequently captured by CALIOP in the altitude range of 0–3 km over ROIs containing or adjacent to the ocean. The regional-averaged





peaks of the stratified FoO for CM aerosols are generally located at 0.1–0.2 km altitude in the near-surface layer. DM aerosols mostly occur below 4 km, and the peaks of the stratified FoO are generally located within the near-surface to 0.4 km altitude,
with higher frequencies in NC, NEA, SA and the ME, with maxima of 9.2%, 15.6%, 12.4% and 17.0%, respectively.

In the ROIs other than SD, NEA, SEA and ME, aerosols are mostly dominated by pollution-type aerosols (i.e., PD, PC and smoke). In addition, pollution-type aerosols tend to have larger interannual fluctuations corresponding to larger standard deviations, influenced by anthropogenic activities and the interannual variability of meteorological conditions. In terms of altitude, the peak values of the stratified FoO for PD and PC aerosols are usually located below 2 km. In contrast, the maxima
of regionally stratified FoO for smoke aerosols all occur between 2 km and 4 km. The most dominant role of smoke aerosol is found to appear at about 2 km over CSA. Except for CSA, the FoO with altitude for smoke aerosols shows a distinct single-peak distribution pattern in AMZ, SA, SEA, SC, NC, NEA, EUS and WEU, corresponding to an altitude of 2.6 km (5.6%), 2.5 km (4.3%), 2.5 km (4.7%), 2.7 km (8.1%), 2.7 km (4.4%), 2.5 km (4.3%), 2.6 km (3.4%) and 2.5 km (2.2%), respectively. In the three ROIs located in the dust source area, the maximum values of the stratified FoO for dust aerosols are located at 1.1
km (33.4%), 1.5 km (35.3%) and 1.5 km (18.1%) for SD, ME and NWC, respectively. As for the downstream ROIs of the dust aerosols, the maximum stratified FoO is located at 1.7 km (9.8%) and 2.4 km (8.9%) in NC and SA, respectively. Notably, in the SD region, we find low standard deviations for all aerosol types, indicating that the interannual variability of aerosols in this region does not fluctuate significantly.

### 3.6. Long-term trends in aerosol loading at different altitude regimes

Benefiting from the continuous record of aerosol vertical observations spanning 13 years, this study further investigates the temporal trends of total aerosol and type-dependent aerosol over different altitude regimes, including the whole layer, sub-layers, the PBL and FT. This is expected to provide new insights to further our understanding of the long-term changes in the total global aerosol loading over the last decade or so. Fig. 13 shows the long-term trends in the total column AODs and their partitioned AODs within the PBL and FT for total aerosols and the three main aerosol types (i.e., dust, PD and smoke),
separately, for the period 2007–2019. These trends were calculated for the annual time series at each 2º × 5º (latitude × longitude) grid box by using Sen's slope method. The results show that TAOD experiences a significant decreasing trend in eastern China, EUS, South America, WEU, and the ME, and a significant increasing trend in SA and central Siberia during the period 2007–2019. These significant regional temporal anomalies, both in terms of spatial patterns and magnitude of variability, are confirmed by satellite observations (i.e., MODIS/Aqua, MODIS/Terra, and MISR) and reanalysis products (i.e.,
MERRA-2 and CAMS) (Fig. 14). Of course, these regional trends are also consistent with previous studies (e.g., Klingmüller et al., 2016; David et al., 2018; Che et al., 2019; Jin and Pryor, 2020; Gui et al., 2021). Despite the differences in sampling frequency and spatial resolution between different AOD data sources, the spatial consistency of TAOD trends further confirms the accuracy of CALIOP observations, especially the ability to capture long-term variability.

Further analysis of the trends in TAOD partitioned within the PBL and FT shows that the trend patterns of AODs within
the PBL (TAOD_PBL) and within the FT (TAOD_FT) are consistent with those of TAOD, but the former have greater



magnitude of variation, confirming the dominant driving role of the TAOD_PBL for the total aerosol changes in the whole layer. Also noticeable is a significant reduction in the number of grid points in some regions (e.g., the ME, SA and tropical Pacific) showing significant variability in TAOD_FT compared to TAOD_PBL, which is likely related to the regional differences in vertical transport efficiency (Cui and Carslaw, 2006). Regional differences in the efficiency of the vertical

transport of aerosols from the PBL to FT in different regions will determine any regional inconsistencies in the regularity of the vertical distribution of aerosol concentrations. Moreover, when the aerosols within the PBL are lifted to the FT, they can then be transported over long distances, leading to the existence of greater spatial heterogeneity in TAOD_FT than TAOD_PBL.

For dust aerosols, we can see a remarkable decreasing trend prevalent on a global scale. The strongest decline in DAOD

is in the ME, reaching $-0.1$ decade$^{-1}$. Modulated by meteorological conditions, inactive dust sources in East Asia over the last decade have contributed to a significant decrease in DAOD in NC, with trend values reaching about $-0.05$/decade. Strong but insignificant upward and downward trends can be seen throughout the northern part of Africa. The declining signs of DAOD in the ME and NC do not shift in the PBL and FT. Remarkably, the downward trend of DAOD partitioning in the PBL (DAOD_PBL) in north-central China shifts to DAOD partitioning in the FT (DAOD_FT) in the east-central China. The

differences in the trend patterns of DAOD_PBL and DAOD_FT over the above regions are mainly attributable to the delay effect of the trans-regional transport of dust aerosols (Yang et al., 2017).

Influenced by a combination of anthropogenic emissions and dust aerosols, PDAOD has experienced significant decreases in eastern China and southern South America, and significant increases in the ME and SA. These regional trends are usually dominated by aerosols distributed within the boundary layer, except for eastern China. Aerosols within both the PBL

and FT in eastern China are comparable in their contributions to the PDAOD trends. For smoke aerosols, significant decreases in SAOD occur in SC and the EUS, with similar non-significant decreases also occurring in South America and southern Africa. In contrast, SAOD enhancement occurs in SA and the Indochina Peninsula, but the magnitude of the trend is not significant. Different from DAOD and PDAOD, the trend in SAOD is mainly driven by the aerosols distributed within the FT, especially in SC. This phenomenon is mainly due to the fact that smoke aerosols can be lifted to the FT rapidly under the action

of a vertical atmosphere.

Fig. 15 further shows the global trend distribution of the partitioning of TAOD, DAOD, PDAOD and SAOD within the altitude layer at every 2-km interval. The results reveal that the decline in TAOD in eastern China, the ME and EUS can be mainly attributed to 0–2 km and 2–4 km, while the increase in SA attributes to 0–2 km. Influenced by the continuous reduction in dust aerosol emissions from dust source areas in the last decade, the intensity of the trans-Pacific transport of dust aerosols

in East Asia has significantly decreased, which has directly led to a significant weakening of dust aerosol loading within 2–10 km altitude over the North Pacific region. In the ME, the reduction in DAOD remains largely consistent across the four altitude layers within 0–8 km. In contrast, the decreasing trend in the layer-specific PDAOD in eastern China occurs mainly within the altitudes of 0–2 km and 2–4 km. In addition, we find that the weakening of TAOD over the North Pacific region is not only attributable to the decreased DAOD, but also contributes significantly to the reduced PD due to the continued diminishing of




anthropogenic aerosol emissions in China (Zheng et al., 2018). Overall, the stratified trend pattern of SAOD at 0–2 km and 2–
4 km remains largely consistent. However, regionally, the overall decline in SAOD in the EUS (SA) can be mainly attributed
to 0–2 km (2–4 km). The decline in smoke aerosols over most of the land directly leads to a significant reduction in SAOD in
the upper atmosphere over the ocean.

The percentage trends (% decade$^{-1}$) in TAOD, DAOD, PDAOD and SAOD over the globe, land, ocean, and 12 ROIs for
the different altitude regimes are summarized in Fig. 16. These altitude-related trends were calculated from regionally averaged
annual time series. Overall, we can see a 4.4% decade$^{-1}$ decrease ($P < 0.05$) in TAOD during 2007–2019. This significant
reduction in global TAOD contributes to the weakening of TAOD, to varying degrees, over both land (−6.1% decade$^{-1}$, $P <$
0.1) and ocean (−3.3% decade$^{-1}$, $P < 0.05$). In terms of altitude, the reduction in column TAOD is attributable to both
TAOD_PBLH and TAOD_FT globally, over land and over ocean, but the former is the more dominant driver. In addition, this
reduction in TAOD is also found simultaneously within all partitioned altitude layers. Among the 12 ROIs, column TAOD
shows a significant decrease ($P < 0.05$) in NEA, NC, SC, ME, WEU, EUS and AMZ, with trend values reaching −19.9%,
−31.3%, −37.3%, −21.0%, −15.5%, −29.6% and −16.9% decade$^{-1}$, respectively. These significant regional trends are present
in both the PBL and FT, in addition to the FT in WEU. In NEA, NC, SC, EUS and AMZ, the decreasing sign of TAOD can
extend all the way from 0–2 km to 8–10 km. However, we find completely reversed trend signs in the altitude range of 0–4
km and above 4 km in WEU, SEA and NWC. Considering that aerosols play different climatic roles at different altitudes, these
findings highlight the importance of exploring the stratified AOD trends. The regional differences in trends within the lower
and upper atmosphere may be related to the high-altitude transport of aerosols.

For dust aerosols, which are mainly driven by meteorology, a non-significant decline in column DAOD is found globally,
over land, and over ocean, and this decline can spread to different altitude regimes. Regionally, the negative DAOD trends in
different altitude regimes are seen simultaneously in NEA, NC, SC and ME. For PD aerosols, PDAOD is found to decrease
by 5.3% ($P < 0.05$), 4.4% and 7.2% ($P < 0.05$) decade$^{-1}$ for the globe, land and ocean, respectively. We also see consistent
negative trend signs in all altitude regimes, albeit of different magnitudes. Regionally, column PDAOD shows a consistent
negative trend in NEA (−38.5 % decade$^{-1}$, $P < 0.05$), NC (−38.1 % decade$^{-1}$, $P < 0.05$), SC (−47.7 % decade$^{-1}$, $P < 0.05$),
while in SA and ME the trend values are completely reversed. These regional trends do not change within the PBL and FT.
Also, the same consistent regional trend sign is seen for the altitude stratification within 0–6 km over the above ROIs. For
smoke aerosols, we find that SAOD experiences a 13.3% decade$^{-1}$ ($P < 0.05$) decline, which is the result of a simultaneous
decline in SAOD over land (−14.7% decade$^{-1}$, $P < 0.1$) and ocean (−10.9% decade$^{-1}$, $P < 0.05$). We do not find a shift in the
sign of these regional trends in the different altitude regimes. Consistent with DAOD and PDAOD, the trend in SAOD at
different altitude regimes is also negative in sign over the NEA, NC and SC regions. However, a positive SAOD trend sign
does extend through different altitude regimes in the SA and ME regions, except for stratification above 6 km.





### 3.7. Effects of meteorological conditions

To explain the trends in total aerosol loading and different types of aerosol loading derived from CALIOP, we examine their relationship with the interannual variability of three key meteorological factors (PPT, VSM and WS). These meteorological factors have been demonstrated to be closely linked to the processes of emission, dispersion, transport and deposition of AAs. The correlation coefficients ($R$) were thus calculated between CALIOP-retrieved TAOD and these meteorological factors from 2007 to 2019. Also, the relationships between meteorology and DAOD and SAOD are examined. As shown in Fig. 17, the most prominent region where TAOD is highly correlated with PPT and WS is Indonesia and its extension into the equatorial central Pacific (ECP). In the ECP region, TAOD is positively correlated with WS and negatively correlated with PPT, indicating that WS and PPT regulate the interannual variability of TAOD mainly by affecting the process of emission and wet deposition of sea-salt aerosols. This implies that the significant decrease in TAOD observed in the ECP region during 2007–2019 (see Fig. 13a) is mainly attributable to the significant increase in PPT (~30%–60% decade$^{-1}$) and significant decrease in WS (~40%–80% decade$^{-1}$) in the region (Fig. 18). In addition, TAOD is positively correlated with WS over the seas of the Southern Hemisphere, implying a dominant driving role of WS for marine aerosols in this region. In contrast, the interannual variability of land aerosols, especially dust and smoke aerosols, is closely related to the variability of PPT and VSM, while the role of WS is not prominent. This can be explained by the fact that PPT mainly modifies surface conditions (e.g., roughness and vegetation cover) in arid or semi-arid areas by affecting VSM, which in turn affects the intensity of dust emissions. For instance, in the ME, a significant increase in PPT (30%–50% decade$^{-1}$) and a slight increase in VSM are responsible for the decreased TAOD over the region (Fig. 18 and Fig. 13a), especially its dust component (i.e., DAOD) (Fig. 13b).

PPT-driven changes in VSM can also alter atmospheric humidity conditions by affecting surface evaporation, which further modifies the frequency of biomass-burning events when induced by high temperatures. This is further confirmed by the dominant negative correlation between TAOD or SAOD and PPT and VSM in typical biomass-burning areas (e.g., the AMZ, central Siberia, SEA, and western United States). Therefore, the observed decrease in TAOD in central Siberia may be related to the decrease in VSM. Reduced VSM can increase the fire risk by changing vegetation conditions as well as atmospheric moisture conditions. However, the changes in these meteorological factors are not really sufficient to explain the decrease in TAOD in eastern China, as we did not detect significant trends in WS and PPT throughout eastern China, except for the observed increase in VSM to varying degrees in north-central China and SC. The increase in VSM (unfavorable to fire occurrence) in SC may be partially responsible for the decrease in regional SAOD. In SA, which is also a region dominated by anthropogenic aerosols, TAOD shows a negative spatial correlation with PPT. Nevertheless, we see a non-significant decreasing trend in PPT only. The above results suggest that a slight decrease in PPT (by affecting wet deposition) may also further enhance regional aerosol pollution levels in the context of increasing anthropogenic aerosol emissions in SA.

CALIOP's unique vertical observation advantage gives us the opportunity to explore the effect of the interannual variability of meteorological factors on TAOD_PBL and TAOD_FT trends. Next, benefitting from this, we focus on a key





question: do the interannual variations in TAOD_PBL and TAOD_FT obey the same relationship between TAOD and
meteorological factors? For this purpose, we further distributed TAOD within the PBL and FT and performed the same spatial
correlation analysis, as shown in Figs. S12 and S13. The results show that the spatial patterns of the *R* between the
meteorological factors and TAOD_PBL remain basically the same as those of TAOD. In terms of the magnitude of the
correlation, the *R* between the meteorological factors and TAOD_PBL becomes more robust compared to the column TAOD
(Fig. S12). In contrast, the distribution patterns of the *R* between meteorological factors and TAOD_FT are quite different
from those of TAOD_PBL, along with a significant weakening of the strength of the correlations, and the signs of *R* are even
reversed in most regions (Fig. S13). The above results indicate that surface meteorological elements affect the interannual
variation in column TAOD mainly by influencing the TAOD_PBL. However, most previous studies on the formation
mechanisms of regional air pollution were based on the relationship between column TAOD (as a key proxy for atmospheric
pollution) and meteorological factors. Therefore, our study suggests that future aerosol-related research should use the variable
TAOD_PBL, which is closely related to near-surface pollutant concentrations, to more realistically elucidate the mechanisms
underlying the influence of meteorological factors on the interannual variability of regional air pollution.

## 4. Conclusions and implications

The vertical distribution of different types of tropospheric aerosols can influence the Earth's climate system through
varying radiative effects. Based on CALIOP Version 4.2 monthly gridded aerosol extinction profiles, averaged over both
daytime and nighttime from 2007 to 2019, this paper comprehensively examines the 3D climatological distribution of AECs
for total aerosol and its sub-types (dust, PD and smoke) in terms of different altitude regimes, including the whole layer,
stratified layers, PBL, and FT. The contribution of TAOD through its partitioning within these different altitude regimes to
column TAOD is also quantified, globally and over 12 ROIs. Then, the FoOs of the seven aerosol sub-types are quantified
using the detection samples of layer aerosols included in the CALIOP Level-3 products to examine the differences in the
vertical distribution of type-dependent FoOs over the globe, land, ocean, and 12 ROIs. On this basis, this study further evaluates
the long-term trends in TAOD and its sub-types, and focuses on elucidating the effects and contributions of long-term changes
in TAOD_PBL and TAOD_FT on TAOD trends. Finally, the effects of three key meteorological drivers on the interannual
variability of TAOD, DAOD, and SAOD partitioned within the whole layer, PBL, and FT are preliminarily explored by
performing spatial correlation analysis.

Over the ocean, enhanced TAEC (0.1 km$^{-1}$) is mostly distributed below 1 km, while the vertical distribution of TAEC
over land varies significantly at regional scales. The hotspots of land TAEC are generally located in the lower levels of dust
source areas and areas dominated by anthropogenic emissions, which show a decreasing trend with altitude. For dust aerosols,
the enhancement of DAEC occurs mainly in the lower atmosphere (below 3 km) inside the dust source and its adjacent region.
In contrast, dust aerosols emitted from the source are mixed with downstream anthropogenic aerosols driven by the
atmospheric circulation, directly contributing several PDAEC hotspots (e.g., SA and eastern China) over areas of intense
anthropogenic activity along the dust transport path. However, the enhanced PDAEC is often limited to below 3 km as it is





constrained by the near-surface emission sources and vertical motion. For smoke aerosols, the vertical distribution of SAEC has a distinct regional dependence, with enhancements occurring mainly in several typical biomass burning regions, including the AMZ, CSA, SEA and central Siberia, where the maximum lifting altitude of smoke plumes exceeds 4–6 km.

When TAOD is partitioned into the PBL and FT through the PBLH data from MERRA-2, we find that the PBL and FT contribute 61.86% and 38.13%, respectively, of the global TAOD during 2007–2019. For DAOD and PDAOD, these partitioning ratios are 49.48% and 50.51%, and 52.22% and 47.79%. In contrast, very different partitioning ratios (23.68% and 76.32% for the PBL and FT, respectively) are seen in SAOD, influenced by the regional vertical transport efficiency, aerosol properties and meteorological conditions. Moreover, a more detailed stratified-layer partitioning suggested the contribution of

the stratified-layer TAOD within 0–2 km, 2–4 km, 4–6 km, 6–8 km, 8–10 km and 10–12 km to TAOD is 70.82% (87.52%), 23.59% (9.41%), 4.82% (2.28%), 0.53% (0.51%), 0.18% (0.21%) and 0.06% (0.07%) over land (ocean), respectively. The ocean–land difference in the contribution implies that continental particles—particularly dust and smoke—are the aerosol types transported the most efficiently to the mid–high levels (~30%, above 2 km), whereas marine particles mostly stay in the marine low levels. Regionally, in all 12 ROIs, more than 50% of the aerosol loading is located in the lower troposphere in the

0–2 km altitude range, while the contribution is less than 2% in altitudes above 6 km. Quantifying the climatology of the FoO of aerosol types categorized by CALIOP shows that, influenced by anthropogenic, dust and smoke aerosol outflows from land, most of the sea areas near land encounter enhanced FoO, while in other sea areas far from land the aerosol FoO is significantly weakened, mainly controlled by sea-salt aerosol. To be specific, on a global average, we found a multi-year averaged whole-layer FoO for all aerosol types of 4.45%, with contributions of 1.79%, 0.86%, 0.30%, 0.05%, 0.57%, 0.38% and 0.50% for

CM, dust, PC, CC, PD, smoke and DM aerosols, respectively. The 3D distribution of FoO indicates that the vertical distribution pattern of stratified-layer FoO for all aerosol types with altitude remains highly consistent with that of TAEC.

      The resulting trend analyses from 2007 to 2019 show that CALIOP-derived TAOD experiences a significant decreasing trend in eastern China, EUS, WEU, the ME, and South America, and a significant increasing trend in SA and central Siberia during this period. It should be noted that, although CALIOP aerosol extinction and/or TAOD have often in previous studies

been reported as underestimated, especially in the high levels, due to detection limitations, our estimated CALIOP TAOD trend pattern over the globe is comparable to the trend obtained from other multi-sensor satellite measurements and reanalysis products. Furthermore, the significant trend patterns of TAOD over most regions except the ECP are largely consistent with TAOD_PBL and TAOD_FT, albeit the former has a larger magnitude of variation. This suggests that the dominant driving role of the TAOD_PBL for column TAOD changes. For dust aerosols, the strongest negative trends are detected over the ME

and NC. The declining signs of DAOD in the ME and NC do not shift in the PBL and FT. Remarkably, the downward trend in DAOD within the PBL in north-central China shifts to DAOD within the FT in the east-central China, which can mainly be attributed to the delay effect of the trans-regional transport of dust aerosols. PDAOD has experienced significant decreases in eastern China and southern South America, and significant increases in the ME and SA. These regional trends are usually dominated by aerosols distributed within the PBL, except for eastern China. For smoke aerosol, significant decreases in SAOD



occur in SC and the EUS. Different from DAOD and PDAOD, the deceased SAOD trend is mainly driven by the aerosols distributed within the FT, especially in SC.

  We attempt to interpret these significant regional trends by linking the interannual variability/trends in aerosol loading to changes in meteorological drivers. Our correlation analysis shows that the interannual variability of TAOD, DAOD and SAOD can be related to variations in PPT, VSM and WS in the particular regions. The positive TAOD trend over the ECP is mainly

attributable to the significant increase in PPT and significant decrease in WS, which emphasizes that WS and PPT regulate the interannual variability of TAOD mainly by affecting the process of emission and wet deposition of sea-salt aerosols. In contrast, the interannual variability of land aerosols, especially dust and smoke, is closely related to the variability of PPT and VSM, while the role of WS is not prominent. For the ME, the negative TAOD/DAOD trends seem to be closely related to increased PPT and VSM. This is because PPT not only impacts dust transport via wet deposition, but also alters dust emissions through

VSM. Moreover, we have also identified regions where PPT and VSM are correlated with the TAOD/SAOD over typical biomass-burning regions. As a result, the increase in TAOD/SAOD detected by CALIOP in central Siberia is inextricably linked to the decrease in PPT-driven VSM. However, for the hotspots of anthropogenic aerosol pollution (i.e., SA and NC), no significant relationship between these meteorological factors and regional TAOD was found during the study period, except for a slight negative correlation with PPT in SA. By performing additional correlation analysis between meteorological factors

and TAOD_PBL and TAOD_DT, it is shown that these significant regional aerosol–meteorology correlations become more robust within the PBL, while weakening significantly or even reversing within the FT. This suggests that TAOD_PBL should be a more suitable indicator than the previously preferred column TAOD to explore the relationship between the interannual variability of aerosol loadings and changes in meteorological drivers, since usually our selected meteorological parameters are based in the near-surface layer.

To conclude, this study provides a comprehensive picture of the 3D structure, distribution, and trends of global and regional tropospheric type-dependent aerosols and their meteorological drivers. The findings may be crucial for reducing uncertainties in the estimation of aerosol direct effects and for further understanding the interannual variability/trend of the global total aerosol loading. It also emphasizes the need to better constrain the AOD within different altitude regimes in models. Finally, the results presented in this study have potential implications not only for quantifying the radiative effect of aerosols,

but also for investigating pollutant transport, modeling air quality, and understanding the effects of aerosols on atmospheric dynamics, cloud fields, and PPT production.

**Data availability.** CALIOP Level-3 gridded aerosol optical profile products and the aerosol and meteorological fields (WS and PBLH data) from MERRA-2 reanalysis are available at https://search.earthdata.nasa.gov/. CAMS global monthly AOD field from ECMWF is available through the Copernicus atmosphere data store

(https://ads.atmosphere.copernicus.eu/cdsapp#!/search?type=dataset). Satellite-retrieved monthly gridded AOD product (MODIS/Terra, MODIS/Aqua, and MISR) are obtained from the Giovanni website (https://giovanni.gsfc.nasa.gov/giovanni/).



GLDAS VSM dataset is available at https://ldas.gsfc.nasa.gov/gldas. GPCP PPT data is obtained from https://psl.noaa.gov/data/gridded/data.gpcp.html.

**Author contributions.** HC and KG designed the study. KG performed the data analysis with contributions from all coauthors;
KG prepared and drafted the paper with help from HC, YZ, HZ, and WY; LL, LZ, HW, YW, and XZ provided constructive suggestions on this study.

**Competing Interests.** The authors declare that they have no conflict of interest.

**Acknowledgements.** NASA's CALIOP/CALIPSO team is gratefully acknowledged for enabling the CALIPSO Level-3 gridded aerosol optical profile data publicly accessible and for their working to improve the aerosol vertical retrieval and
aerosol type classification algorithms.

**Financial support.** This research has been supported by the National Science Fund for Distinguished Young Scholars (No. 41825011), the National Natural Science Foundation of China project (No. 42030608), and the Basic Research Fund of CAMS (No. 2021Y001).

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

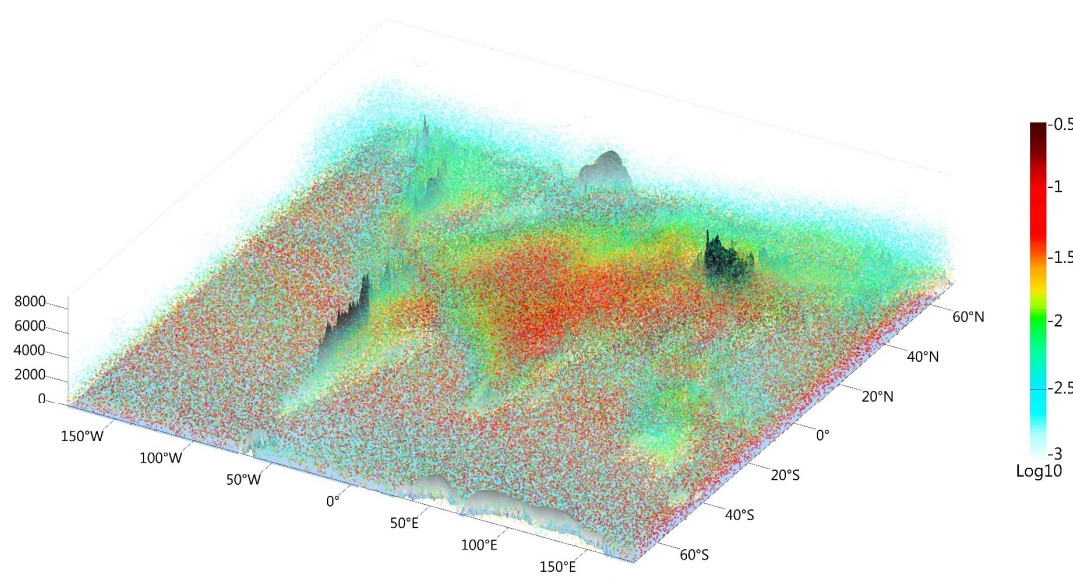

**Figure 1. 3D particle map of the annual total (i.e., all aerosol types) aerosol extinction coefficient (AEC; unit: km$^{-1}$) averaged over 13 years (2007–2019). Note that the color scale has been converted to $\text{Log}_{10}^{\text{AEC}}$ for better visualization, corresponding to a minimum AEC of 0.001 and a maximum AEC of about 0.32.**





**Figure 2. As in Figure 1 but for type-dependent aerosols: (a) dust; (b) polluted dust (PD); and (c) smoke.**





**Figure 3. Spatial distributions of multi-year averaged layer-specific (integrated at 2-km intervals from the surface to 12 km) (a) total AOD (TAOD), (b) dust AOD (DAOD), (c) PD AOD (PDAOD), and (d) smoke AOD (SAOD).**




## Planetary boundary layer    Free troposphere

**Figure 4. Spatial distributions of multi-year averaged AODs for (a) total aerosols, (b) dust, (c) PD, and (d) smoke in the planetary boundary layer (left-hand panels) and free troposphere (right-hand panels). The global average is labeled in the lower-left of each panel.**


**Figure 5. Regional average vertical profiles of multi-year averaged AEC for total aerosols over the 12 regions of interest (ROIs). The shading represents the multi-year average standard deviation. The rectangular boxes marked with different colors on the top map represent the boundaries of each ROI (see Table S1 for more information).**




**Figure 6. Regional-averaged vertical profiles of multi-year averaged AEC for type-dependent aerosols (dust, PD and smoke) over the 12 ROIs. The shading in each subpanel represents the multi-year average standard deviation.**




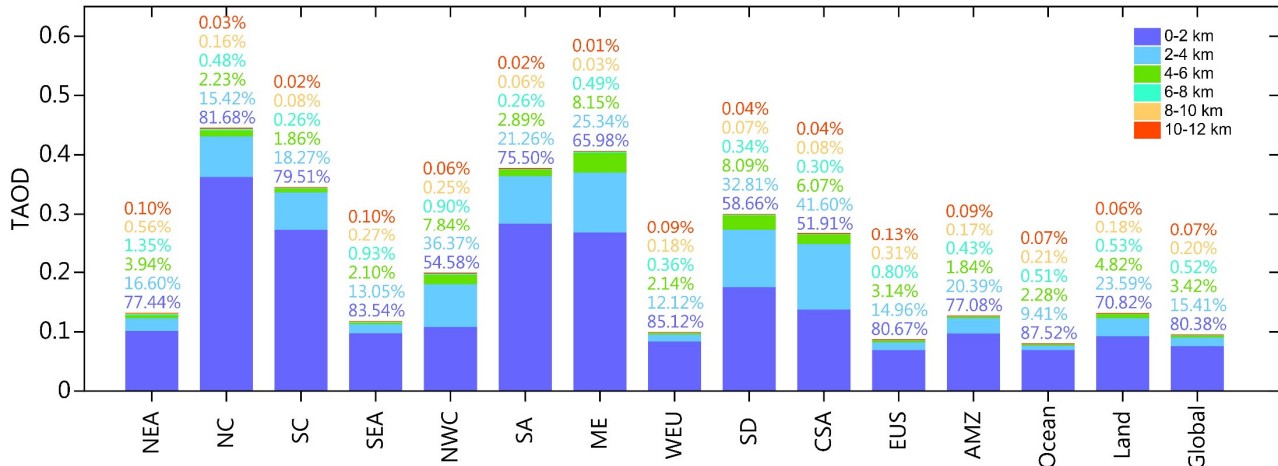

**Figure 7. Multi-year regional-average layer-specific total AODs (integrated at 2-km intervals between 0 and 12 km) for the globe, land, ocean, and 12 ROIs. The colored numbers above each bar represent the relative proportions (%) of the layer-specific AODs to the total AODs.**





**Figure 8. Multi-year regional averages of the (a) the AODs and (b) frequency-of-occurrence (FoO) for different types of aerosols over the globe, land, ocean, and 12 ROIs. The number next to each bar represents the regional total FoO or total AOD.**






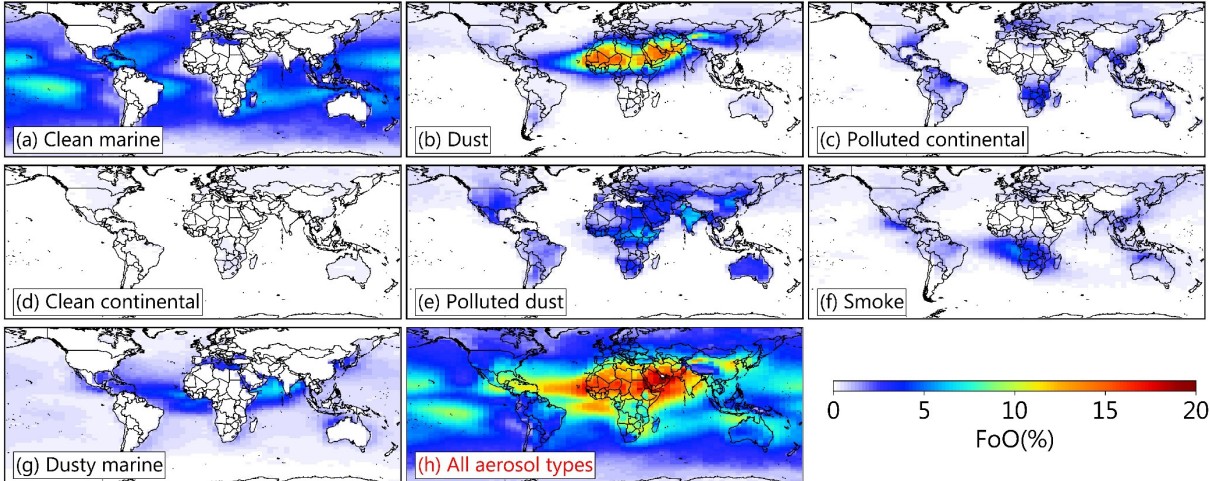

**Figure 9. Spatial distribution of the multi-year average total columnar FoO for (a–g) CALIOP-classified aerosol subtypes and (h) the sum of all aerosol subtypes.**



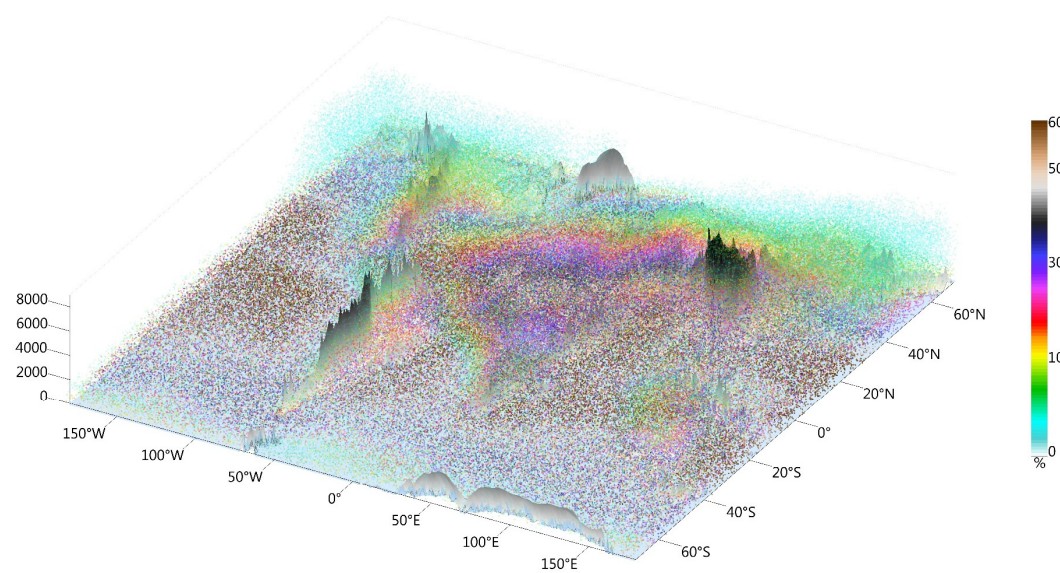


Figure 10. 3D particle map of the annual FoO for all aerosol types averaged over 13 years (2007–2019).



**Figure 11. As in Figure 10 but for type-dependent aerosols: (a) dust; (b) PD; (c) smoke; (d) polluted continental (PC); (e) dusty marine (DM); (f) clean marine (CM).**





**Figure 12. Regional-average vertical profiles of multi-year average FoO for type-dependent aerosols over the 12 ROIs. The shading in each sub-panel represents the multi-year average standard deviation. Note that the altitudes corresponding to the maximum FoO**
**for different types of aerosols are also indicated in each panel.**



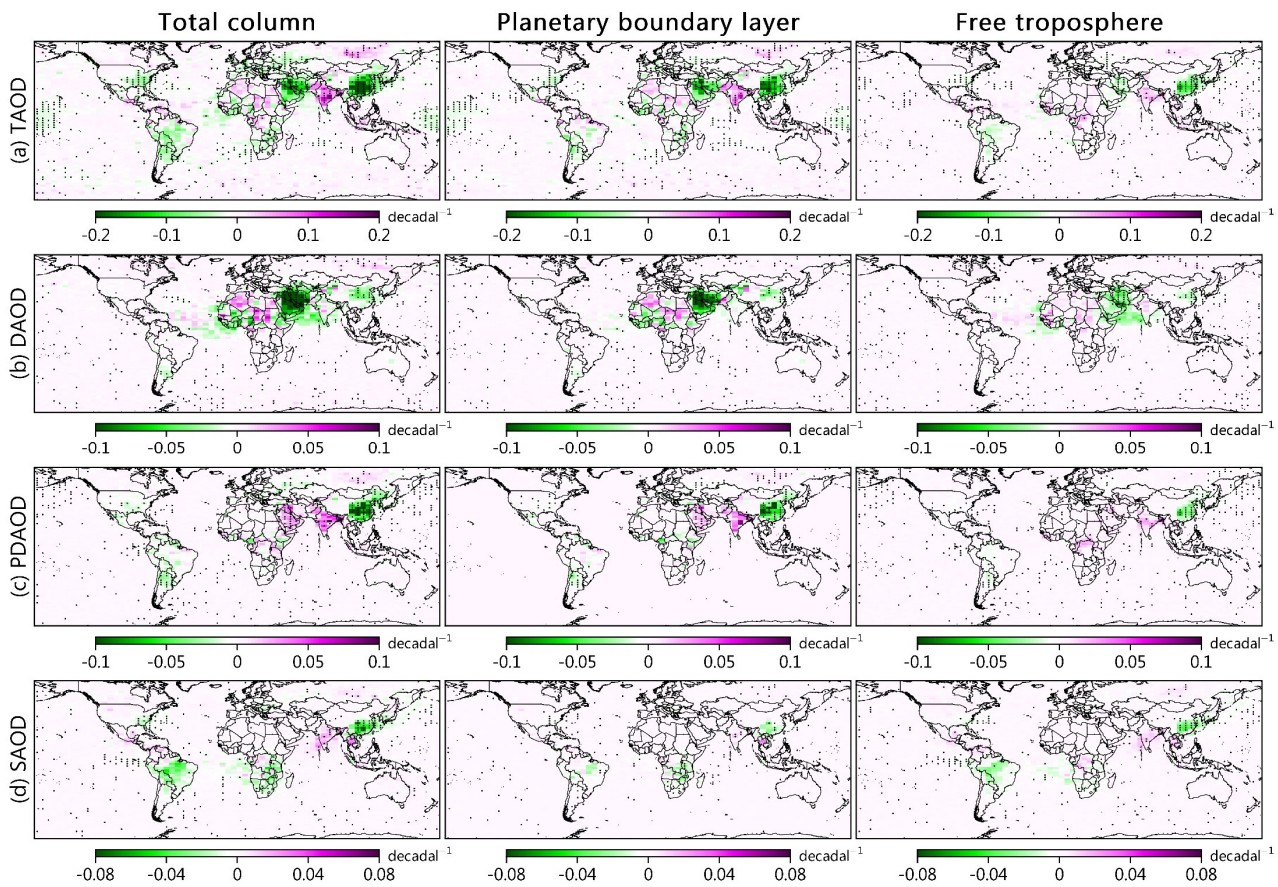

**Figure 13. Spatial distributions of annual trends (unit: decade⁻¹) in (a) TAOD, (b) DAOD, (c) PDAOD, and (d) SAOD in the total column (left-hand panels), PBL (middle panels), and FT (right-hand panels), during the period 2007–2019. The grid points with trend values that are statistically significant at the 95% confidence level are marked with black "×" symbols.**


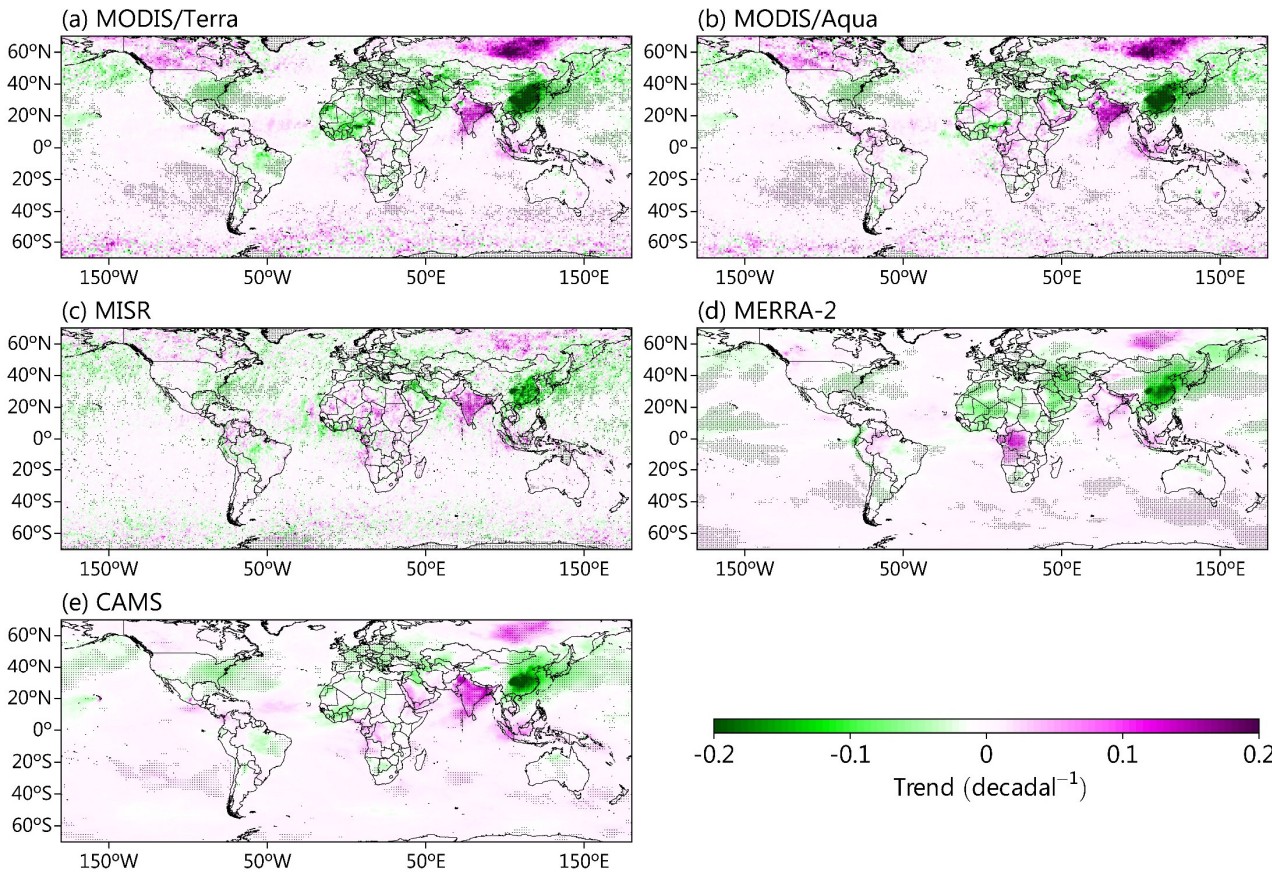

**Figure 14. Spatial distributions of annual trends (unit: decade$^{-1}$) in TAOD calculated from the annual mean time series from (a) MODIS/Terra, (b) MODIS/Aqua, (c) MISR, (d) MERRA-2, and (f) CAMS during the period 2007–2019. The grid points with trend values that are statistically significant at the 95% confidence level are marked with black "×" symbols.**






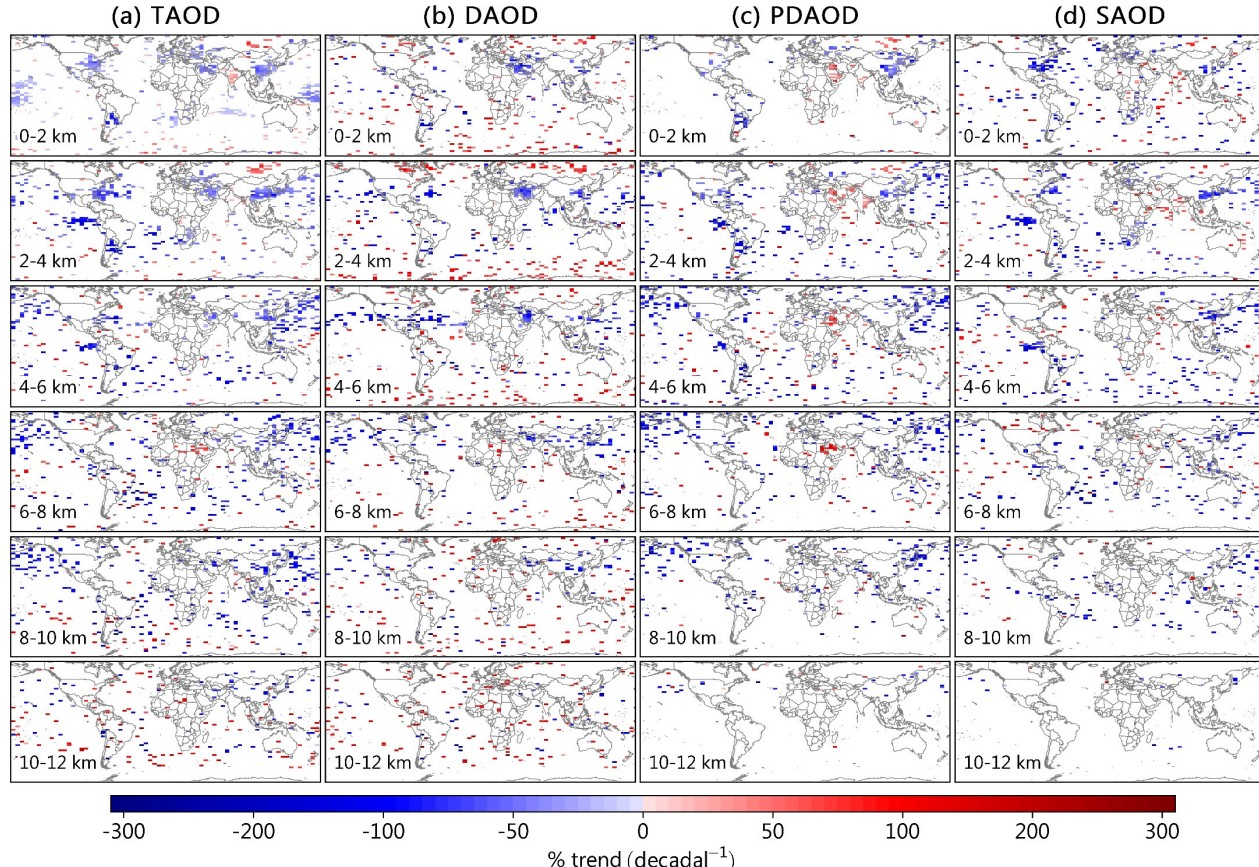

**Figure 15. Spatial distributions of annual trends (unit: % decadal⁻¹) in layer-specific (a) TAOD, (b) DAOD, (c) PDAOD, and SAOD, during the period 2007–2019. Only grid points with trend values that are statistically significant at the 95% confidence level are shown. For visualization purposes, all estimated trend values are given as percentage decadal changes:** $\frac{10 \times Slope}{\bar{y}} \times 100.0\%$ **, where Slope indicates Sen's slope and** $\bar{y}$ **indicates the annual average.**





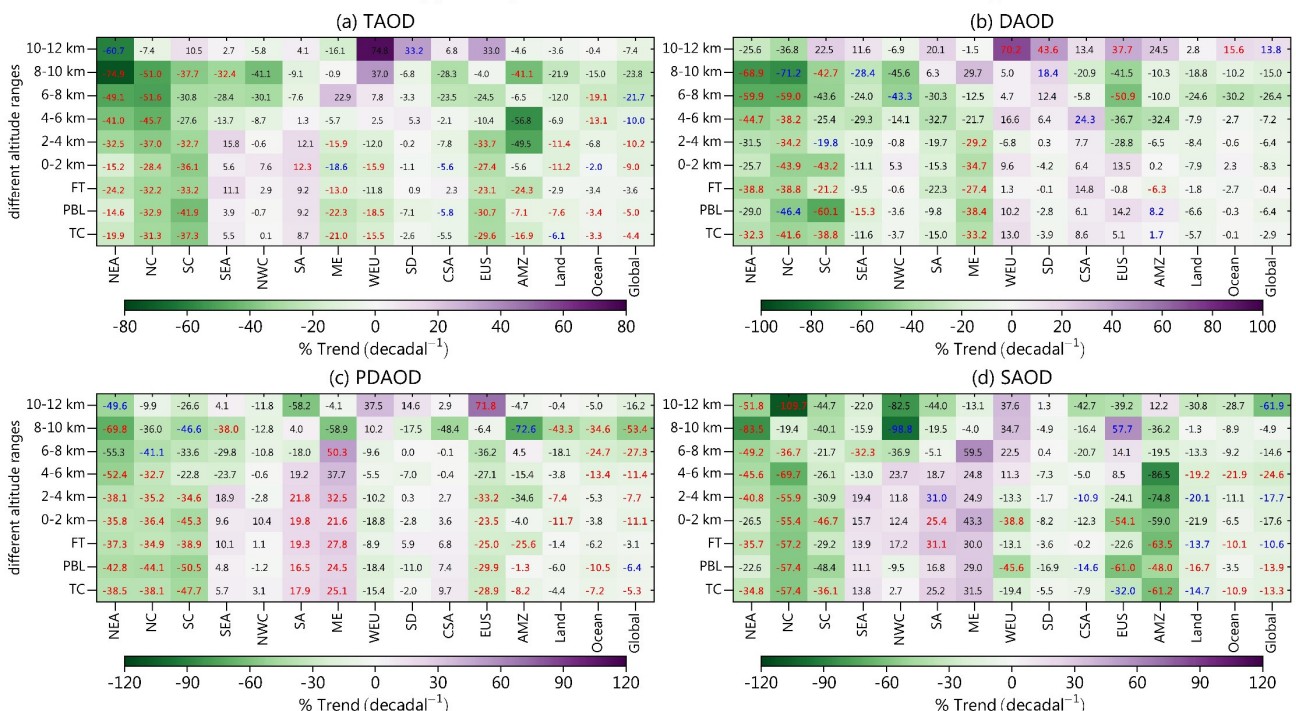

**Figure 16. Global and regional percentage trends (unit: % decade$^{-1}$) in TAOD (a) and type-dependent AODs, including (b) DAOD,**
**(c) PDAOD, and (d) SAOD at different altitude ranges, during the period 2007–2019. The colored squares represent the magnitude**
**of the trend value; the numbers represent the corresponding trend value; and the numbers with red and blue fonts denote that the**
**trend values are statistically significant at the 95% and 90% significance levels, respectively. TC: Total column; PBL: Planetary**
**boundary layer; FT: Free troposphere.**

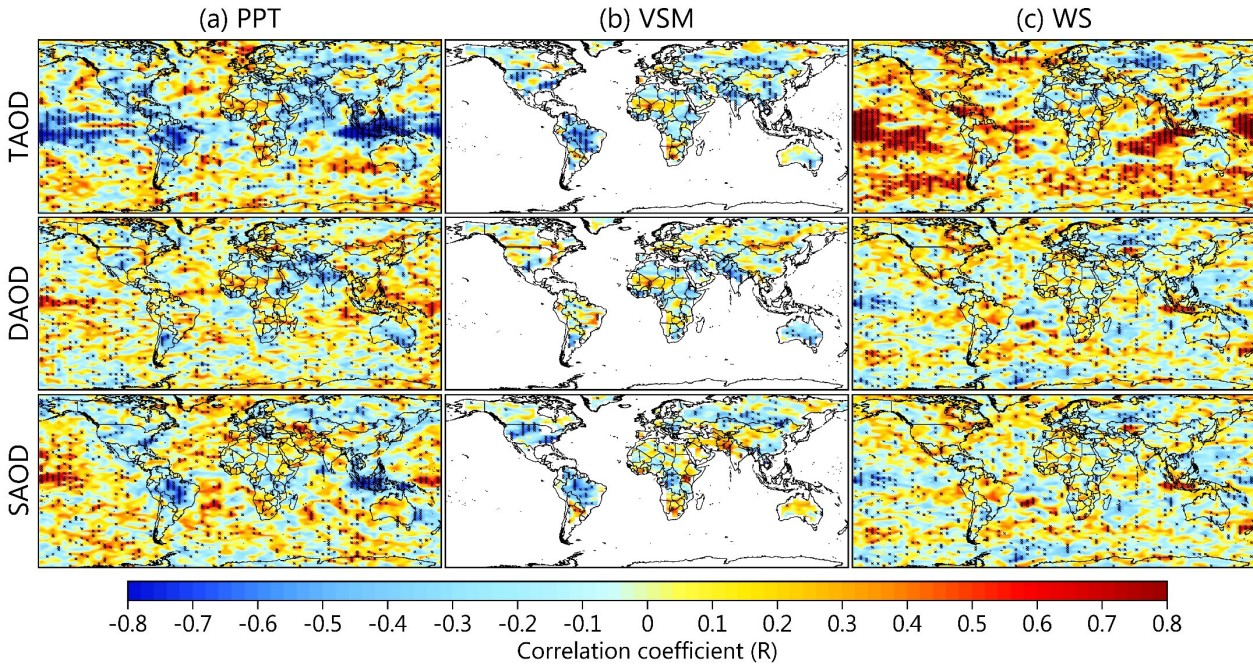


**Figure 17. Spatial distributions of the correlation coefficients (*R*) for TAOD, DAOD, and SAOD versus (a) precipitation (PPT), (b) volumetric soil moisture (VSM), and (c) wind speed at 10 m (WS) for 2007–2019. The grid points with *R* values that are statistically significant at the 95% confidence level are marked with black "×" symbols.**


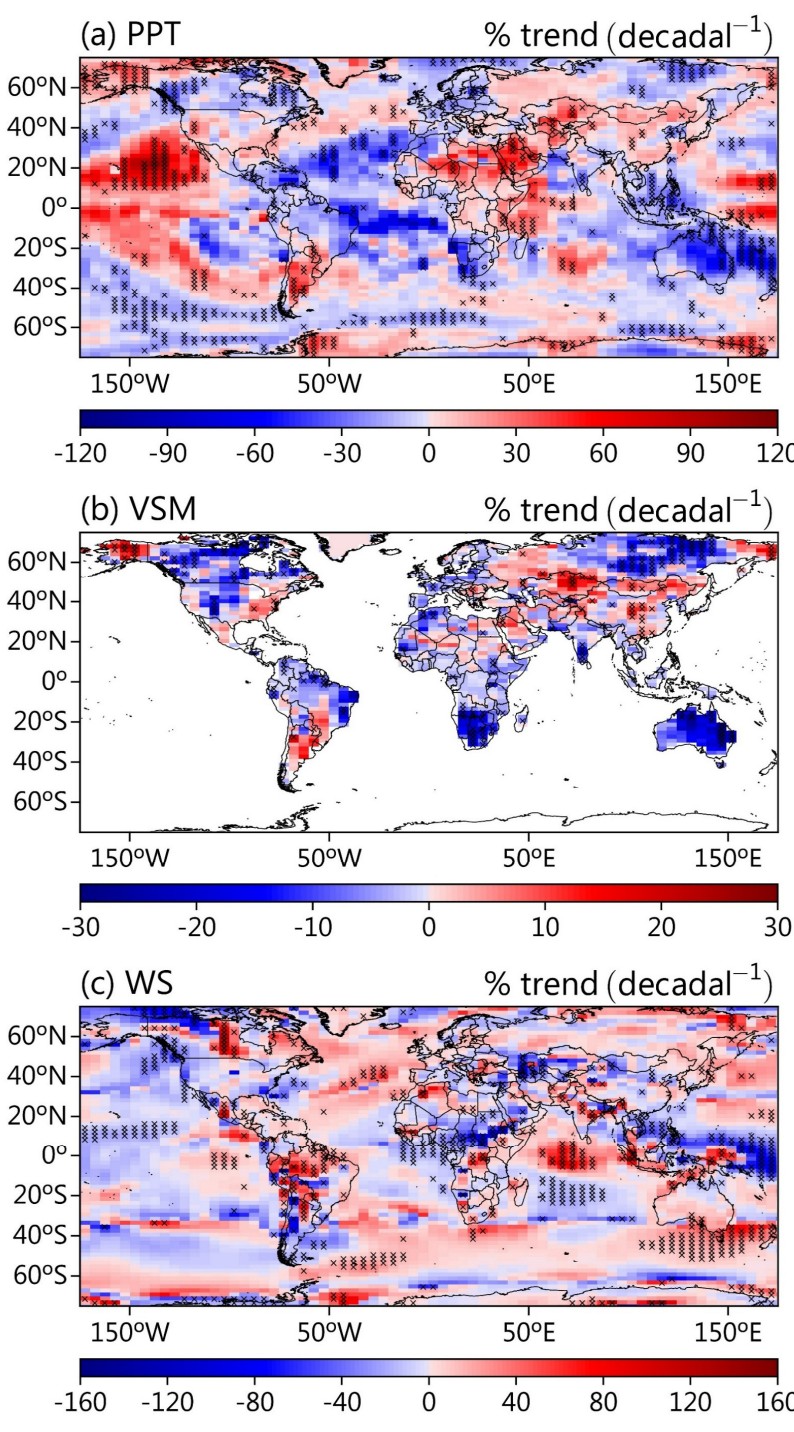

**Figure 18. Spatial distributions of annual trends (unit: % decade⁻¹) in (a) precipitation (PPT), (b) volumetric soil moisture (VSM), and (c) wind speed at 10 m (WS) during the period 2007–2019. The grid points with trend values that are statistically significant at the 95% confidence level are marked with black "×" symbols.**