# Peer review of "Three-dimensional climatology, trends and meteorological drivers of global and regional tropospheric type-dependent aerosols: Insights from 13 years (2007–2019) of CALIOP observations"

_Atmospheric Chemistry and Physics, 2021_

## Author Comment (AC1)

Thanks very much for the time and efforts that you have put into reviewing the previous version of the manuscript. We really appreciate all your comments and suggestions that have enabled us to improve the manuscript. Attached is a point-to-point response to reviewer's comments. We have studied comments carefully and have made correction which we hope meet with approval. Revised portion are marked in red in the revised paper.

**Reviewer #1:**

1. This study displayed the spatiotemporal and vertical distribution characteristics of AOD with the CALIOP data as well as AOD retrievals from other satellite sensors and reanalysis AOD data at a global scale. It also analyzed the associations between AOD and meteorological factors. The detailed analyses by region and by aerosol type contributed to our understating of the variation trends of atmospheric aerosols.

**Response:** Thank you for your positive comments on our work. We have revised it in accordance with your comments or suggestions. For detailed revisions, please refer to the following sections.

2. In the introduction section, the author provided a detailed introduction on the importance of studying the atmospheric aerosols; however, the summary of previous studies and findings is lacking, making it hard to evaluate the contribution of this study to this field.

**Response:** Following the suggestions of the reviewers, we have revised the introduction to highlight the contributions of our study. Some previous studies have been added, and the main revisions are as follows:

**Lines 89-101 in the revised paper:**

"Although these efforts have significantly improved our understanding of the vertical distribution of aerosols, only a limited number of studies have utilized CALIOP observations to examine the partitioning of the total AOD (TAOD) and the AODs due to different aerosol subtypes within different altitude regimes. Bourgeois et al. (2018) reported the amount of TAOD present in the PBL and FT on a global scale using CALIOP data together with the PBL heights (PBLHs) obtained from the ERA-Interim (European Centre for Medium-range Weather Forecasts Re-Analysis-Interim) archive. Shi et al. (2020) investigated the characteristics of aerosol in the residual layer and its effects on the surface PM2.5 over China using ten-year CALIOP data. Vinjamuri et al. (2020) explored the vertical distribution of smoke aerosols against the PBL and average injection height of smoke aerosols over the upper Indo-Gangetic Plain using CALIOP attenuated backscatter lidar profile. In addition to providing total aerosol extinction profiles, the CALIOP classifies the total aerosol profiles into different source types with different physical characteristics using an aerosol classification algorithm. Therefore, the availability of long-term (>10 years) continuous observations from CALIOP makes it possible to deepen our understanding of the interannual variations and trends of tropospheric type-dependent aerosol loading partitioned within different altitude regimes and their meteorological drivers."

Compared with existing studies, the main contribution of our study is to obtain the climatology of TAOD and type-dependent AOD partitioning within different altitude regimes and its long-term trends using CALIOP observations. In addition, another recommended highlight is the examination of the relationships between the interannual variability of aerosol loading within the boundary layer and free troposphere and the meteorological drivers.

**References:**

- Vinjamuri, K. S., Mhawish, A., Banerjee, T. and Sorek-hamer, M.: Vertical distribution of smoke aerosols over upper Indo-Gangetic Plain, Environ. Pollut., 113377, doi:10.1016/j.envpol.2019.113377, 2019.
- Shi, Y., Liu, B., Chen, S., Gong, W., Ma, Y., Zhang, M., Jin, S. and Jin, Y.: Characteristics of aerosol within the nocturnal residual layer and its effects on surface PM2.5 over China, Atmos. Environ., 117841, doi:10.1016/j.atmosenv.2020.117841, 2020.
- Bourgeois, Q., Ekman, A. M. L., Renard, J.-B., Krejci, R., Devasthale, A., Bender, F. A.-M., Riipinen, I., Berthet, G. and Tackett, J. L.: How much of the global aerosol optical depth is found in the boundary layer and free troposphere?, Atmos. Chem. Phys., 18(10), 7709–7720, doi:10.5194/acp-18-7709-2018, 2018.

3. In the data and methods section, the method used for data assimilation may not be appropriate. The VSM data (1-degree spatial resolution) and WS data (0.5 degree \* 0.625 degree) were at higher spatial resolutions than the CALIOP data (2 degree \* 5 degree). Thus, the VSM and WS data should be averaged to match the CALIOP grid, but the author used bilinearly interpolation to assimilate the data. The PBLH data from MERRA-2 were also assimilated by bilinearly interpolation, which may lead to systemic bias.

**Response:** We are very grateful to the reviewers for their scientific advice, which has been fully considered in the revised version. Based on the reviewer's suggestion, we modified the method in the original manuscript from a bilinear interpolation to a simple averaging method. The result is that we made updates to the following five figures in the main text (i.e., Figures 4, 13, 16-18). Overall, the change in resampling method has a negligible effect on the results. For example, the distribution patterns of climatology of TAOD and AODs due to different aerosol types partitioned within the PBL and FT varied little (see Fig.4). Similarly, their trend distributions also changed little, both in terms of intensity and spatial patterns (see Figs.13 and 16). Moreover, the spatial distribution of the correlation coefficients between the partitioned AODs within the PBL and FT and meteorological drivers remain unchanged (see Fig.17). Also, the trend patterns of meteorological drivers have not changed (see Fig. 18). Nevertheless, we have double-checked the corresponding descriptions in the text.

Finally, we choose the average method recommended by the reviewers to present the results of this work. Consequently, "bilinearly interpolated" in the original manuscript was changed to "aggregated".

---

## Author Comment (AC2)

Thanks very much for the time and efforts that you have put into reviewing the previous version of the manuscript. We really appreciate all your comments and suggestions that have enabled us to improve the manuscript. Attached is a point-to-point response to reviewer's comments. We have studied comments carefully and have made correction which we hope meet with approval. Revised portion are marked in red in the revised paper.

**Reviewer #2:**

1. This study investigated the three-dimensional (3D) climatological distribution of tropospheric type-dependent aerosols using the CALIOP globally gridded aerosol extinction data during 2007–2019. This study focused on filling the limited spatial and temporal coverage of ground-based Lidar, and have significantly improved the comprehensive understanding of the vertical distribution of aerosols and their climatic effects. Moreover, the regional tropospheric type-dependent aerosols and their association with meteorological factors for more than 10 years was provided, which is helpful for pollutant transport, modeling air quality research. Overall, the manuscript is well written, has a complete structure, and has many impressive visualizations presented. However, I think, this paper need some revision before final acceptation. Below are some of my detail comments to the submitted manuscript that I hope the authors can use to improve their study.

**Response:** Thank you for your positive comments on our work. We have revised it in accordance with your comments or suggestions. For detailed revisions, please refer to the following sections.

2. In this study, trends in CALIOP L3 monthly AODs are compared against several other datasets. Although CALIOP is independent in relation to the other four datasets, these datasets are not independent of each other. Two datasets (MERRA-2 and CAMS) are not independent as they incorporate information from both MISR and MODIS retrievals. In addition, there are temporal sampling differences between observations, such as differences in satellite crossing times. Therefore, it is important for the authors to add more details to this when describing the data and to discuss, as much as possible, the possible influence of these factors on the results. Finally, I'm also interested in whether CALIOP has good synergy with the other datasets over the 12 ROIs.

**Response:** The reviewers' suggestions are greatly appreciated, and we have added the following more detailed descriptions of the different AOD data sources, including the relationships between them.

Lines 210-214 in the revised paper:

*"It should be noted that although the CALIOP-derived AOD is independent, other AOD datasets are not completely independent of each other. For example, MERRA-2 assimilates the aerosol information from MODIS and MISR, and CAMS assimilates MODIS. However, given the fact that each dataset has its own strengths and limitations in characterizing global and regional aerosol loadings. Thus, by comparing CALIOP*

*with other AOD data, we expect to reveal consistencies of CALIOP with respect to other products.”*

Regarding consistency between different AOD data sources. We calculated the region-averaged AODs over 12 ROIs using CALIOP and the other five datasets (i.e., MISR, MODIS/Terra, MODIS/Aqua, MERRA-2, and CAMS) and compared the consistency between their interannual variations, and the results are shown below.

[Figure]

**Figure S12.** Interannual variations in regional AOD averaged over the 12 regions of interest (ROIs). Note that the colored numbers inserted at the top of each panel represent the correlation coefficients (*R*) between CALIPSO and the other five datasets. Numbers labeled with * and ** represent *R* values above the 90% and 95% significance levels from two-tailed Student's t-tests, respectively.

As we see, the CALIOP-derived AOD has a good agreement with other datasets at the interannual variability scale. Therefore, the following description was also added to the revised version.

Lines 541-543 in the revised paper:

*“In addition to long-term trends, CALIOP also shows good consistency with other AOD*

*products in terms of year-to-year variation over all ROIs except for NWC and CSA, albeit with slight differences in magnitude (Fig. S12)".*

3. There is no doubt that the AOD (especially the AOD within the boundary layer) has a significant diurnal variability. Although CALIOP is the only satellite-based lidar currently operating for more than 10 years, its relatively long cycle time and incomplete global coverage make it difficult to assess the impact of diurnal variations of aerosols on the results of this study. Therefore, it is suggested here that the authors look forward to future work, especially using other observational instruments that enable all-weather observations (e.g., CATS: Measuring Clouds and Aerosols from the International Space Station) to explore the diurnal variability of aerosol vertical distribution and partitioning at different altitudes.

**Response:**
The reviewers' suggestions are greatly appreciated and we have added some outlooks for future work, as follows:

Lines 730-733 in the revised paper:
*"Although CALIOP provides early afternoon and morning observations, two temporal points and a 16d repeat cycle are insufficient to evaluate the diurnal variations of aerosol properties within different altitude regimes. Thus, the observations of a near-full diurnal cycle of aerosol properties [e.g., the Cloud-Aerosol Transport System (CATS) onboard the International Space Station] (Lee et al., 2019; Cheng et al., 2020) should be incorporated to address this limitation in future work. To conclude, this study provides a comprehensive picture of the 3D structure, distribution, and trends of global and regional tropospheric type-dependent aerosols and their meteorological drivers. The findings may be crucial to reduce uncertainties in the estimation of aerosol direct effects and to better constrain the AOD within different altitude regimes in models. Finally, the results presented in this study have potential implications not only for understanding the interannual variability/trend of the global total aerosol loading, but also for examining the role of meteorological conditions in modifying the interannual variability of different types of aerosols."*

References:
Lee, L., Zhang, J., Reid, J. S., and Yorks, J. E.: Investigation of CATS aerosol products and application toward global diurnal variation of aerosols, Atmos. Chem. Phys., 19, 12687–12707, https://doi.org/10.5194/acp-19-12687-2019, 2019.
Cheng, Y., Dai, T., Li, J., and Shi, G.: Measurement Report: Determination of aerosol vertical features on different timescales over East Asia based on CATS aerosol products, Atmos. Chem. Phys., 20, 15307–15322, https://doi.org/10.5194/acp-20-15307-2020, 2020.

4. The conclusion section is too lengthy and the authors should refine the results of this study and highlight the notable contributions.

**Response:** The reviewers' suggestions are greatly appreciated and we have streamlined the conclusion section to highlight the important findings of this study. Please refer to the revised manuscript for details.

5. Line 123, 99.99 km$^{-1}$ ? Please check it.

**Response:** Thanks for your thoughtful suggestion. It is 99.99 km$^{-1}$. Corrected.

6. Line 126, I did not find the eighth QC from this technical literature, please confirm.

**Response:** Thanks for your thoughtful suggestion. The eighth QC description has been removed.

7. Line 154, Reduce to some extent? Expressions are too absolute.

**Response:** Thanks for your thoughtful suggestion. We have made revisions accordingly.

8. Line 156, Remove the extra "-".

**Response:** Thanks for your thoughtful suggestion. Corrected.

9. Line 234-235, This wording is slightly unclear. Consider rewording as either "we required that at least 60% of the data in each annual time series be valid before the trend calculations could be performed" or "we needed to obtain at least 60% of the valid data in each annual time series before performing the trend calculations" depending on your meaning.

**Response:** Thanks for your thoughtful suggestion. The original sentence has been changed to *"we required that at least 60% of the data in each annual time series be valid before the trend calculations could be performed."*

10. Line 255, the author can briefly explain their climate effects, such as absorption?

**Response:** Regarding the climatic effects of aerosols, we think we have summarized them in detail in the introduction.

**11.** Line 255-259, This part of the description should be moved to the method data introduction section.

**Response:** As suggested by the reviewers, these expressions have been moved to the introduction.

12. Line 329-330 and Line 340-342, Add reference.

**Response:** The following references were included.

1. Fromm, M., Tupper, A., Rosenfeld, D., Servranckx, R., and McRae, R.: Violent pyro-convective storm devastates Australia's capital and pollutes the stratosphere, Geophys. Res. Lett., 33, L05815, https://doi.org/10.1029/2005GL025161, 2006.
2. Generoso, S., Bey, I., Labonne, M., and Bréon, F.-M.: Aerosol ver- tical distribution in dust outflow over the Atlantic: Comparisons between GEOS-Chem and Cloud-

Aerosol Lidar and Infrared Pathfinder Satellite Observation (CALIPSO), J. Geophys. Res., 113, D24209, https://doi.org/10.1029/2008JD010154, 2008.

3. Yumimoto, K., Eguchi, K., Uno, I., Takemura, T., Liu, Z., Shimizu, A. and Sugimoto, N.: An elevated large-scale dust veil from the Taklimakan Desert: Intercontinental transport and three-dimensional structure as captured by CALIPSO and regional and global models, Atmos. Chem. Phys., 9(21), 8545–8558, doi:10.5194/acp-9-8545-2009, 2009.

4. Höpner, F., Bender, F. A.-M., Ekman, A. M. L., Praveen, P. S., Bosch, C., Ogren, J. A., Andersson, A., Gustafsson, Ö., and Ramanathan, V.: Vertical profiles of optical and microphysical particle properties above the northern Indian Ocean during CARDEX 2012, Atmos. Chem. Phys., 16, 1045–1064, https://doi.org/10.5194/acp-16-1045-2016, 2016.

13. Line 349, the word "studies" could be changed as "study".
**Response:** Thanks for your thoughtful suggestion. Corrected.

14. Line 357, the word "shows" could be changed as "show".
**Response:** Thanks for your thoughtful suggestion. Corrected.

15. Line 360, "reaches a maximum value" or "reaches its maximum".
**Response:** Thanks for your thoughtful suggestion. Corrected.

16. Line 384, the words "aerosol" could be changed as "aerosols".
**Response:** Thanks for your thoughtful suggestion. Corrected.

17. Line 401, "1 %" could be changed as "1.0 %", and the decimal number is unified in the text.
**Response:** Thanks for your thoughtful suggestion. Corrected.

18. Line 409, the word "values" can be deleted.
**Response:** Thanks for your thoughtful suggestion. Corrected.

19. Line 472, the word "(i.e., CC and CC)", What are these two CC refer to.
**Response:** Thanks for your thoughtful suggestion. Corrected.